# Learning positional encodings in transformers depends on initialization

## Abstract

The attention mechanism is central to the transformer's ability to capture complex dependencies between tokens of an input sequence. Key to the successful application of the attention mechanism in transformers is its choice of positional encoding (PE). The PE provides essential information that distinguishes the position and order amongst tokens in a sequence. Most prior investigations of PE effects on generalization were tailored to 1D input sequences, such as those presented in natural language, where adjacent tokens (e.g., words) are highly related. In contrast, many real world tasks involve datasets with highly non-trivial positional arrangements, such as datasets organized in multiple spatial dimensions, or datasets for which ground truth positions are not known, such as in biological data. Here we study the importance of learning accurate PE for problems which rely on a non-trivial arrangement of input tokens. Critically, we find that the choice of initialization of a learnable PE greatly influences its ability to learn accurate PEs that lead to enhanced generalization. We empirically demonstrate our findings in three experiments: 1) A 2D relational reasoning task; 2) A nonlinear stochastic network simulation; 3) A real world 3D neuroscience dataset, applying interpretability analyses to verify the learning of accurate PEs. Overall, we find that a learned PE initialized from a small-norm distribution can 1) uncover interpretable PEs that mirror ground truth positions (with respect to isometry) in multiple dimensions, and 2) lead to improved downstream generalization in empirical evaluations. Importantly, choosing an ill-suited PE can be detrimental to both model interpretability and generalization. Together, our results illustrate the feasibility of learning identifiable and interpretable PEs for enhanced generalization.

## 1 Introduction

Transformers commonly use ordered sequences of data, like words in a sentence. The position and order of these words are crucial to their correct interpretation. In transformers, sequences of tokens (e.g., words) are processed in parallel – not sequentially. Thus, to process tokens correctly in their intended sequence, the transformer must encode a notion of position and/or ordering of tokens. This information is encoded in its positional encoding (PE) layer – a model parameter that tags each input token with a unique location. For many common forms of data, such as natural language, text, and audio, the labeling of ground truth positional information is straightforward, since tokens are ordered sequences in 1D. This led to the original design of 1D sinusoidal PEs, which were successfully applied to natural language data, and provided general spatial information about language tokens (rather than data-specific information) (Vaswani et al., 2017). More recent investigations into the role of PE in transformers has led to a proliferation of PE schemes, each specifically designed for 1D text with different properties (Su et al., 2022; Shaw et al., 2018; Vaswani et al., 2017; Raffel et al., 2020; Li et al., 2024; Kazemnejad et al., 2023; Shen et al., 2024; Golovneva et al., 2024; Press et al., 2022). However, many interesting problems require input sequences that are not in 1D (e.g., image datasets; Li et al. (2021)), or where position information is non-trivial or not known (e.g., spatially-embedded biological data). The choice of PE significantly affects the performance of transformer models, even in simple string-based tasks (Kazemnejad et al., 2023; Ruoss et al., 2023; McLeish et al., 2024). Thus,

understanding how to disambiguate and learn ideal position information from data directly would likely provide improved performance while affording increased model flexibility.

If learning the optimal PE for a task can enhance downstream generalization performance, what strategies can we use to achieve this? Recent work in deep learning theory suggests various parameterizations of simple neural network models, such as weight initializations, can greatly influence their learned internal representations (Woodworth et al., 2020; Chizat et al., 2020). In particular, those studies found that weight initializations in neural networks from large-norm distributions (e.g., a normal distribution with a large standard deviation) learned random, high-dimensional representations that would "memorize" input-output relations. This learning regime is commonly-referred to as the *lazy* learning or *neural tangent kernel* (NTK) regime, as it fails to learn a structured representation of the task or input. In contrast, neural networks that were initialized from a small-norm distribution (e.g., $\mathcal{N}(0, \sigma)$ for small $\sigma$) tend to learn structured representations that accurately reflected the organization of input features and were robust to noise. This is referred to as the *rich* or *feature* learning regime (Woodworth et al., 2020; Chizat et al., 2020). (We note that choosing the initialization rank can also induce rich versus lazy learning; Liu et al. (2024)). Although this theoretical framework was initially developed for simple neural networks (e.g., feed-forward networks with few hidden layers), the insights drawn from it should apply to various model architectures, including transformers (Zhang et al., 2024; Kunin et al., 2024). Given the recent interest in studying the impact of PEs on generalization, we aimed to evaluate whether the norm of PE initialization would influence the ability to properly learn a structured and accurate PE that would enhance generalization and interpretability.

Here we studied how the initialization of learnable PEs in transformers influence representation learning and downstream generalization. We focused on problems containing sequences with nontrivial positioning and ordering, comparing the generalization performance of models with learned PEs to other common PE schemes (such as absolute and relative PEs). We also examined the interpretability of the learned PEs. *Our primary aim was to evaluate the hypothesis that learnable PEs in the feature rich learning regime would produce interpretable position information that mirrored ground truth knowledge and improve generalization performance.* We tested this hypothesis in three experiments: 1) A 2D relational reasoning task called the Latin Squares Task (LST), which is analogous to simplified Sudoku; 2) Masked prediction of a nonlinear stochastic network simulation with spatially-embedded nodes; 3) a real-world neuroscience dataset, where the task is to predict masked brain activity in 3D from spatially and heterogeneously distributed brain regions. Overall, we found that when the PE is appropriately initialized with a small norm, learnable PEs can uncover ground truth PEs, which lead to improved downstream generalization in both datasets. Note that the notion of *ground truth* positions is task-dependent and data-dependent. In some cases, such as in biological datasets, ground truth spatial information may be difficult to know or ambiguous. In this study, we focus on tasks in which either a ground truth is unambiguous (e.g., synthetic tasks) or in which there exists a putative ground truth (e.g., biological data with known properties). These results indicate the importance of PE choice for generalization performance, and provide insights into how to optimally discover PEs for a variety of tasks in which the ground truth PE is nontrivial or not known.

## 1.1 Contributions

We highlight three principal conclusions of this study.

1. Using a 2D relational reasoning task and nonlinear network simulation with known ground truth position information, we demonstrate that the ability to approximate the ground truth PE (with respect to isometry) depends on initializing a PE parameter with a small norm.

2. We demonstrate the generality of this approach to learning nontrivial PEs in a real world 3D neuroscience dataset, where only small-norm initialized PEs learn a PE representation of brain regions that reflect brain network modularity.

3. We demonstrate that in all experiments, learning an accurate PE enhances downstream generalization relative to alternative and commonly-used PEs.

## 2    Model Architecture

For all three experiments, we used a standard encoder-only transformer architecture with four layers and embedding dimension of 160 for the LST, 64 for the stochastic nonlinear network simulation, and 64 for the fMRI data (Vaswani et al., 2017). The primary transformer manipulation was the choice of PE, and includes a mix of absolute, relative, and learnable PEs (described in ensuing sections). The formulations and definitions of common PEs are detailed in the Appendix A.1. For each experiment, we trained on 15 seeds. For simplicity of analyzing attention maps, we trained models with only a single attention head (fully-connected, bidirectional attention unless specified otherwise). However, we have included results for models with multiheaded attention (2 and 4 heads) on the LST task in Fig. A15, which reduced generalization performance. The context window for the model was either 16 tokens long for the LST (given the $4 \times 4$ structure of the LST paradigm), 15 tokens for the NMAR model, or 360 tokens long (for the number of brain regions in the Glasser et al. (2016) brain atlas). We used the Adam optimizer with a learning rate of 0.0001. For comparable analysis, all models were trained for a fixed number of training steps (4000 epochs for the LST; 8000 puzzles per epoch; 50k training steps for human brain data). Results reported in the main text were trained without regularization, given that we were interested in understanding the role of PE initialization in isolation (no dropout, no weight decay). However, for completeness, we include results using weight decay (AdamW, with weight decay=0.1) in the Appendix for the LST task, which yielded qualitatively similar results (Fig. A8, Table A5, Table A6). A single model/seed could be trained (4000 epochs of LST) on one NVIDIA V100 GPU in under 45 minutes.

## 3    Experiment 1: Relational reasoning in the Latin Square Task (LST)

We first evaluated the effect of initialization on learning accurate PEs on the LST, a 2D relational reasoning task that is similar to the game Sudoku (Fig. 1A-D). The LST is a nonverbal relational reasoning task developed in line with the psychological theory of *Relational Complexity* (Birney et al., 2006; Halford et al., 1998). Prior work in humans has demonstrated the reliability of the LST and its relationship to fluid intelligence (Hearne et al., 2020; Birney et al., 2012; Hartung et al., 2022). Each puzzle in the LST involves the presentation of a 4-by-4 grid populated with stimuli (e.g., shapes, numbers, etc.,), blank spaces, and a single target probe location, noted with a question mark. The fundamental rule of the LST is that in a complete puzzle (i.e., there remain no empty squares), *each shape can only appear once in each row and column.* In our setup, the agent's aim is to infer the unknown target stimuli based on the organization of the elements within the LST grid.

The number of relations needed to solve a given LST puzzle can be manipulated by changing the organization of the elements within the grid (Fig 1A-D). For instance, 1-vector puzzles require integration of information across a single row *or* column, while 2-vector problems involve integration across a single row *and* column. 3-vector puzzles require information integration across three rows and/or columns. Importantly, performing the LST task requires positional information of the rows and columns. *Flattening the grid to 1D without preserving the position information would significantly increase the difficulty of the LST task* (e.g., see Fig. 1E). For our LST experiments, we generated 8000 training puzzles, and assigned to 1-, 2-, or 3-vector conditions. We ensured that the similarity between any generated training set puzzle was distinct from the generalization (validation) set of puzzles (i.e., the *Jaccard dissimilarity* $> 0.8$ for a test puzzle to any individual training puzzle). Moreover, simpler neural network architectures, such as vanilla MLPs and bidirectional LSTMs, could not generalize well across this split (Fig. A16).

### 3.1    PE initialization influences downstream generalization in the LST

Prior work in deep learning theory suggests that choice of weight initialization can influence learned representations and downstream generalization (Jacot et al., 2020; Woodworth et al., 2020; Chizat et al., 2020). In particular, the smaller the norm of the distribution from which the neural network is initialized, the more structured the learned representations will be. We first assessed whether these intuitions would generalize to learnable PEs initialized from different Normal distributions, controlling for the standard deviation (i.e.,

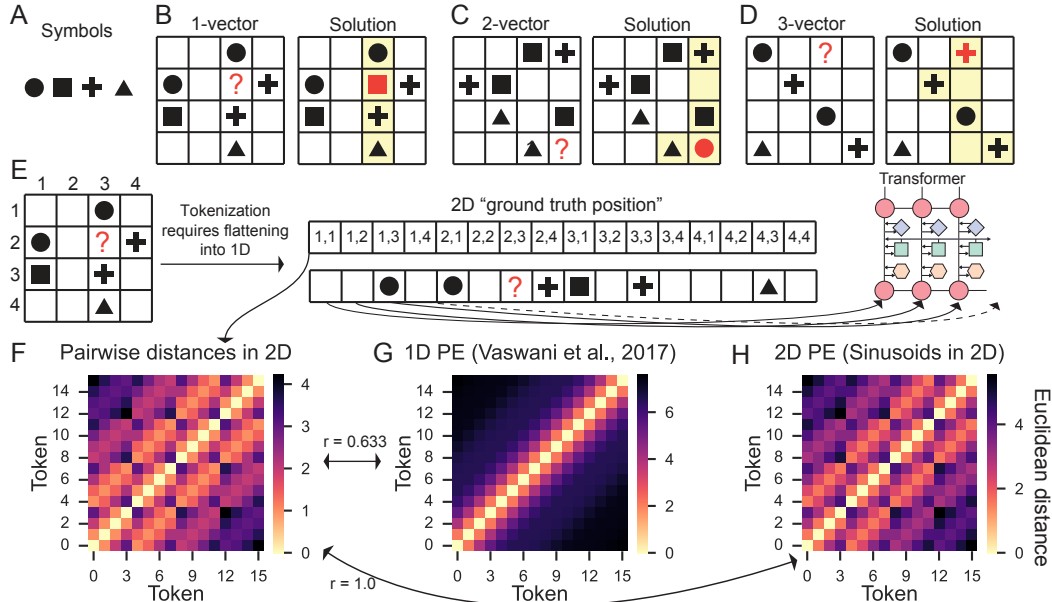

Figure 1: A-D) The Latin Square Task (LST). The LST involves the presentation of a 2-dimensional 4-by-4 grid populated with A) four possible symbols, blank spaces and a single target probe ("?"). The agent's aim is to solve for the target square with the rule that *each shape can only appear once in every row and column.* The reasoning complexity required for an LST puzzle can be manipulated by varying the number of distinct vectors that must be integrated to solve the problem. Examples of B) 1-, C) 2-, and D) 3-vector LST puzzles (left) and their solutions (right). E) Performing the LST is intuitive in 2D. However, when formatting the task sequence for neural networks, the input must be flattened into 1D. The LST is significantly more challenging when row and column information is lost. F) The pairwise distances between token positions according to rows and columns provides the "ground truth" of how tokens relate to each other in 2D space. *A successful PE would preserve the isometry (i.e., pairwise distance relationships) of the (x,y) grid coordinates.* G) Naively using the 1D sinusoidal PE (Vaswani et al., 2017) would provide incorrect token-wise position information, since it only considers the closeness of tokens in 1D. H) In contrast, recomputing absolute positions in 2D (with sines and cosines in the embedding dimensions) would preserve the position information of a 2D grid, even after flattening the sequence into 1D.

norm). For each token embedding, we initialized a learnable PE parameter from a multivariate Normal distribution, denoted $\mathcal{N}(\mathbf{0}, \mathbf{\Sigma})$, where $\mathbf{\Sigma} = \sigma \mathbf{I}$, and $\mathbf{I}$ denotes the identity matrix scaled by $\sigma$ (Fig. 2A).

We initialized PEs from distributions with $\sigma \in \{0.1, 0.2, 0.3, ..., 2.0\}$. (For each PE initialization, we trained on 15 seeds.) We trained all models for 4000 epochs, and found that all models converged (Fig. 2B). Remarkably, though the choice of $\sigma$ was the only source of variation across model parameterizations, models exhibited a wide range of generalization performance (Fig. 2C). Compared to the default initialization choice of $\sigma = 1$, which had a generalization performance of 0.89 (Table A2), we found that the optimal generalization performance was produced with PEs initialized with small norms (e.g., $\sigma = 0.2$; Acc=0.96; Table A2). Moreover, consistent with the NTK regime, PE's initialized from a distribution with large $\sigma$ converged, but generalized poorly (e.g., $\sigma = 2.0$, Acc=0.38; see also Table A2).

While our results are consistent with the hypothesis that small norm initialized models tend to learn the most generalizable representations, we did observe a slight reduction in generalization performance for PEs initialized at $\sigma = 0.1$ relative to $\sigma = 0.2$. In practice, we found that the poor generalization for very small values of $\sigma$ is due to the behavior of the Adam optimizer. Specifically, vanilla SGD outperformed the generalization ability of models trained with Adam for small values of $\sigma \in \{0.01, 0.05, 0.1\}$ (see Fig. A7). Thus, it is important for practitioners to consider optimizers when encouraging the rich/feature training regime for very small initializations.

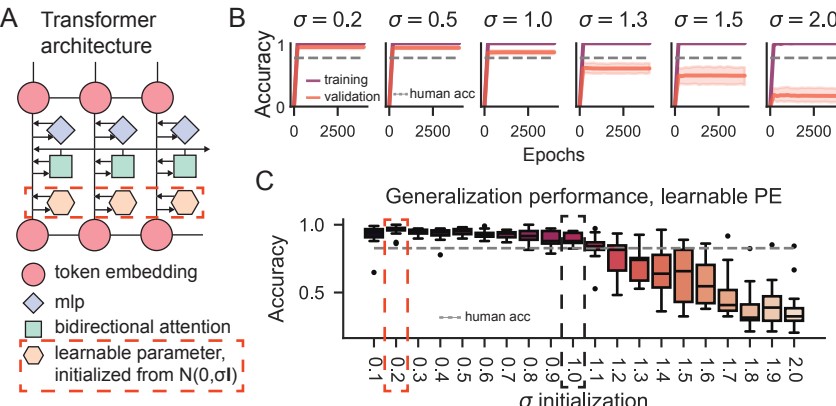

Figure 2: A) Transformer parameterization. We parameterize a learnable PE initialized from different distributions (i.e., $\mathcal{N}(0, \sigma\mathbf{I})$, varying $\sigma$), and study the effect initialization on downstream generalization. B) The training and validation performance across 4000 epochs for examplar initializations. C) Despite the learnability of all initialized models, the choice of $\sigma$ strongly influenced downstream generalization. Boxplots reflect variability across 15 random seeds. (See also Appendix Table A2.)

Table 1: Training and validation performance of common PEs and the `learn-0.2` PE on the LST.

| PE | Validation acc | Validation SD | Training acc | Training SD |
|---|---|---|---|---|
| **2d-fixed (ground truth)** | **0.977** | **0.073** | 1.000 | 0.000 |
| **learn-0.2** | **0.956** | **0.039** | 1.000 | 0.000 |
| 1d-relative | 0.920 | 0.042 | 1.000 | 0.000 |
| random | 0.888 | 0.046 | 1.000 | 0.000 |
| 1d-rope | 0.805 | 0.115 | 1.000 | 0.000 |
| 1d-fixed | 0.781 | 0.185 | 0.999 | 0.000 |
| nope | 0.334 | 0.020 | 0.509 | 0.002 |
| c-nope | 0.314 | 0.042 | 0.559 | 0.102 |

### 3.2 Learnable PEs outperform commonly-used PE schemes in the LST

We next sought to benchmark the optimal learnable PE ($\sigma = 0.2$, `learn-0.2`) relative to other standard PE schemes. These schemes included absolute 1D PE (`1d-fixed`, using sines and cosines; Vaswani et al. (2017)), 1D relative PE (`1d-relative`; Shaw et al. (2018)), and 1D rotary PE (`1d-rope`; Su et al. (2022)). In addition, we also evaluated performance on PE schemes that have been shown to be beneficial for algorithmic and compositional generalization tasks, including no PE with a causal attention mask (`c-nope`) (Kazemnejad et al., 2023) and random PE without a learnable parameter (`random`) (Ruoss et al., 2023). (Note that the `c-nope` model can implicitly learn position information due to the nature of the causal attention mask.) We also included a baseline control model without any specified PE (`nope`), which should not be able to learn the task in a systematic way, due to the permutation invariance of bidirectional attention in the absence of PE. Finally, we included a "ground truth" PE – an absolute 2D PE based on sines and cosines (`2d-fixed`) – to compare how similarly the various PEs produced attention mechanisms to this ground truth model (see Appendix A.1). (Note, that the term "ground truth" applies to any rotation of the absolute 2D PE, or any PE that preserves the pairwise isometry between the row and column information in the 2D LST grid. This could include alternative absolute 2D PEs not based on sinusoidals, in addition to relative PEs that preserve the isometry of the 2D grid information in the LST, as depicted in Fig. 1E,F).)

All models, except for the `nope` and `c-nope` models converged (accuracy after 4000 epochs, `c-nope`=0.56, `nope`=0.51). Poor performance was expected for the `nope` model due to the lack of any explicit or implicit PE information, and `c-nope` models neither learned nor generalized due to the autoregressive and causal structure of its attention mask. Nevertheless, as anticipated, we found the ground truth `2d-fixed` model to exhibit the highest generalization performance (Table 1; Appendix Fig. A10A,D). Remarkably, the

next highest performing model was the `learn-0.2` model ($\sigma = 0.2$), followed by the `1d-relative`, `random`, `1d-rope`, and `1d-fixed` PE models, respectively. While we did not find a robust statistical difference between the ground truth `2d-fixed` model (97.7%) and the `learn-0.2` model (95.6%) (`2d-fixed` vs. `learn-0.2`, t-test, $t(13) = 0.96$, $p = 0.35$), there was a significant difference between the `learn-0.2` model with the next highest-performing model (`1d-relative`) ($t(13) = 2.51$, $p = 0.03$). In addition, we assessed the interaction of PE initialization with weight decay (AdamW), a commonly-used L2 regularization scheme (Loshchilov and Hutter, 2019). We empirically found that the combination of low initialization and weight decay improved generalization; performance was virtually indistinguishable from the ground truth `2d-fixed` PE (both models generalized with 99% accuracy; Table A6; Fig. A8). Finally when performing a perturbation analysis, where we systematically injected noise to the token embeddings and evaluated downstream performance, we found that models that were most robust to noise were those initialized from a low-norm distribution (Fig. A11). Overall, these findings corroborate theoretical findings in deep learning, suggesting their applicability to learning rich transformer PEs in structured reasoning tasks.

### 3.3 Learnable PE models discover ground truth attention maps and positions

Next, we sought to understand how different PE initializations influenced the learned representations (e.g., attention maps and learned PEs) within the transformer. Since the underlying structure of the LST paradigm is a 2D grid, we used the `2d-fixed` model as the ground truth model. This allowed us to assess how learned attention maps would deviate from attention maps derived from optimal position information. First, we extracted the attention weights for each model, and computed the cosine similarity of attention weights between the ground truth and learnable PE models (Fig. 3A; Table A3). We then correlated the alignment of learned attention maps to the ground truth with generalization performance. We found that the degree of agreement of learned attention maps with the ground truth (`2d-fixed`) predicted improved generalization ($\rho = 0.96$, $p < 0.0001$; Fig. 3B). Importantly, learned PEs with small $\sigma$ tended to learn attention representations that were more aligned with the ground truth. (We also computed the Jensen-Shannon Divergence as a complementary distance measure, finding similar results; Appendix Fig. A9.) In the Appendix, we include comparative analyses of the attention maps of models with common PEs (`1d-fixed`, `1d-relative`, etc.) and compare those to the ground truth model (Fig. A10B-D).

Next, to directly interpret what embeddings the learned PEs converged to during training, we measured the alignment of learned PEs with the ground truth `2d-fixed` PEs (Fig. 3D). This involved estimating the distance (i.e., L2 norm) between the `2d-fixed` PE and the learned PE embedding after an orthogonal Procrustes transform was applied. An orthogonal Procrustes transform was applied to rotate and match embedding dimensions according to maximal similarity since embedding dimensions were arbitrary in the learnable PE models. We found that small-norm initialized PEs could better approximate the PEs from the `2d-fixed` PE scheme (Fig. 3E). Critically, more similar ground truth PE approximation (measured by L2 norm) near perfectly predicted downstream generalization ($\rho = -0.98$, $p < 0.0001$; Fig. 3F). These results indicate that 1) low-norm initializations can discover ground truth PEs and their subsequent attention maps, and 2) these discovered PEs predicted downstream generalization.

## 4 Experiment 2: Masked prediction of a nonlinear stochastic network simulation

In our next experiment, we sought to demonstrate the impact of PE initialization in transformers in learning data representations from a nonlinear, stochastic network simulation. Specifically, the main goal in this experiment was to assess the interpretability of learned representations, rather than generalization. Unlike natural language text, many real-world networks, such as genetic networks, air traffic flows, and brain propagation networks, are inherently spatially embedded in physical or conceptual space. Thus, understanding the positional arrangements of embedded data – e.g., the spatial relation between two genes in DNA, or the functional relation between two brain regions – is of paramount importance when learning representations of that data. We first explore the importance of learning PEs in a toy network simulation embedded into clusters of networks, before subsequently investigating its importance in real-world brain data in the subsequent section.

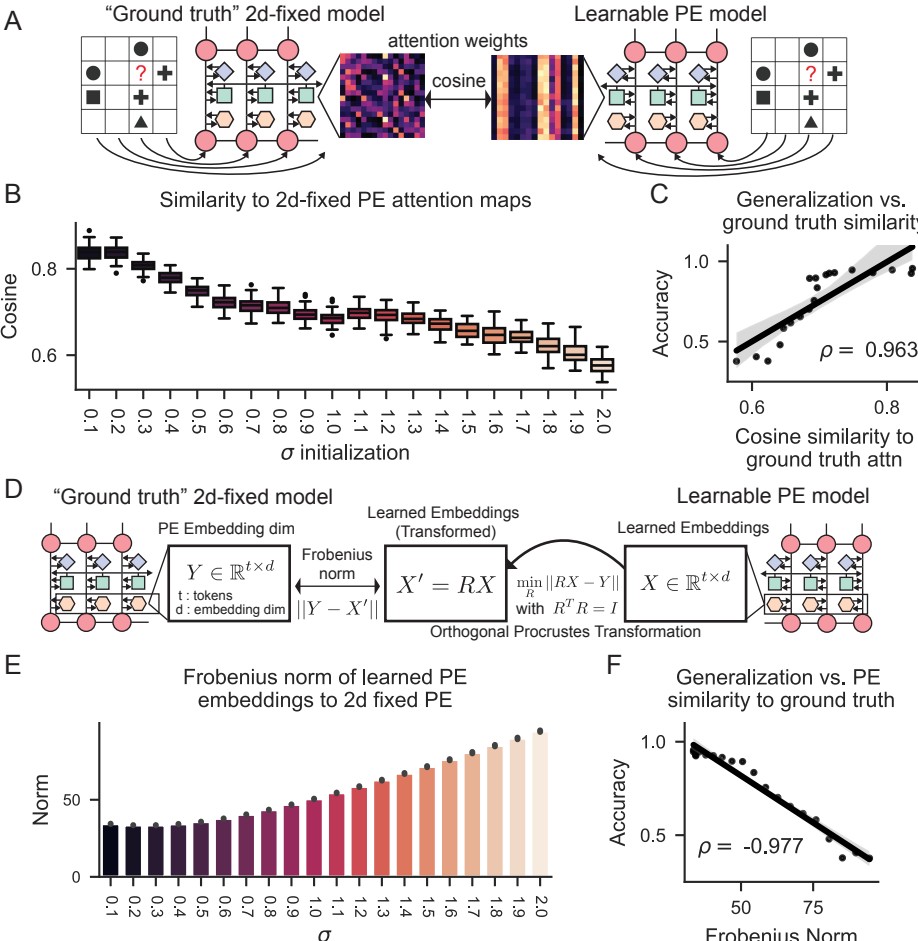

Figure 3: A,B) We compared the attention maps of each learnable PE model to those derived from the "ground truth" `2d-fixed` model. C) A strong rank correlation between generalization performance and the agreement of attention maps to the `2d-fixed` model. D) We directly compared the learned PE embeddings to the ground truth `2d-fixed` PE. Since the embedding dimensions of the learned PE models are random and not indexed in the same way to the `2d-fixed` embedding, we matched their embedding dimensions using an orthogonal Procrustes transform. After matching the dimensions of the PE embeddings, we computed the L2 norm to calculate the distance between the learned PE and the ground truth `2d-fixed` PE. E) The distance (L2 norm) between the learned PEs and `2d-fixed` PE, for every $\sigma$. F) We found a strong relationship between the PE agreement with the ground truth PEs and generalization performance ($\rho = -0.977$). (See also Appendix Table A3.)

## 4.1 Recovering network modules via PE learning in a stochastic network simulation

We implemented a nonlinear multivariate autoregressive (NMAR) model to simulate a system of 15 nodes organized into 3 network clusters (5 nodes per network) (Fig 4A). Each node evolves over time based on a combination of: 1) Autoregressive effects from its own past values ($p = 3$ lags); 2) Strong intra-cluster interactions with other nodes in the same cluster; 3) Weak inter-cluster interactions with nodes in different clusters; 4) Private noise to introduce variability.

The timeseries for node $x_i$ at time $t$ was computed as

$$x_i(t) = \sum_{k=1}^{p} w_{i,k} \cdot x_i(t-k) + \sum_{j \in C_i, j \neq i} \lambda_{ij} \cdot f(x_j(t-1)) + \sum_{j \notin C_i} \eta_{ij} \cdot f(x_j(t-1)) + \epsilon_i(t),$$

where $w_{i,k} \sim \mathcal{U}(0.2, 0.5)$, $C$ refers to the module, $\lambda_{ij} \sim \mathcal{U}(0.02, 0.2)$, $\eta_{ij} \sim \mathcal{U}(0.005, 0.01)$, $\epsilon_i(t) \sim \mathcal{N}(0, 0.2)$, and $f(x) = \sin(x)$.

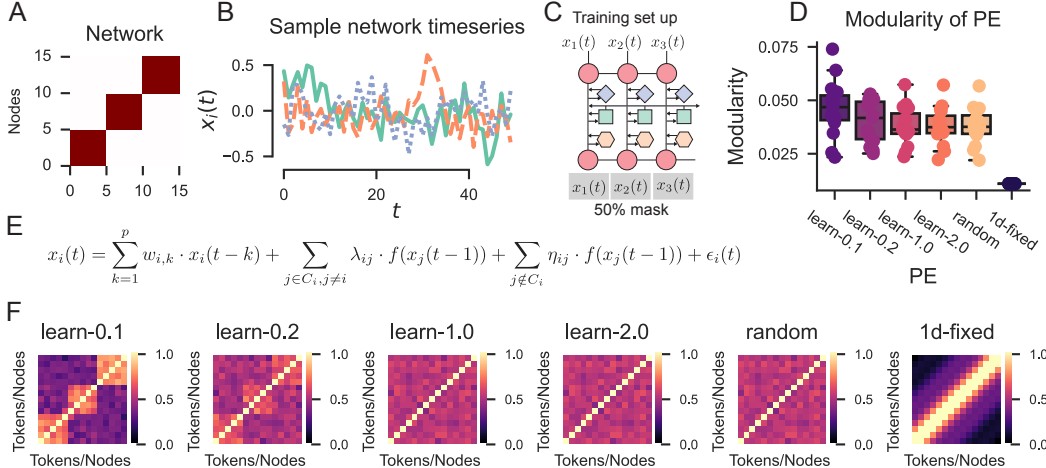

Figure 4: Recovering network structure in learned PEs using a simple nonlinear, stochastic network simulation. A) Network structure with 15 nodes divided into 3 modules, where intra-module nodes interacted more strongly with each other than inter-module nodes. B) Example time series from 3 randomly selected nodes. C) We used a self-supervised training objective, where the model was trained to predict masked data from contemporaneous timepoints. D) The network modularity of PEs, which was measured in relation to the "ground truth" networks, was highest in learnable PE models initialized with small norm (`learn-0.1`) (see Appendix A.3 for calculation). Boxplots are sorted by mean modularity in descending order. E) The equation governing the timeseries generation for node $x_i(t)$ (see text for description of parameters). F) The cosine similarity of the PEs for each pair of tokens, across all models. We can observe that learnable PE models with small-norm initializations (particularly `learn-0.1`) can recover the ground truth network structure, while others cannot. (For additional data on training and testing convergence, see Fig. A17.)

Models were simulated for 20k time points. The training objective for the transformer was to predict contemporaneous activity of the 15 nodes using masked inputs (mask-level=50%) (Fig. 4C). We found that learnable PE models with a small-norm were the models that were most capable of learning the ground truth network organization (see visualizations in Fig. 4F, and compare to Fig. 4A). Statistically, when computing the network modularity of the learned PEs with respect to the ground truth network structure, we found that models initialized with learnable PEs with $\sigma = 0.1$ learned positions with the highest modularity (Fig. 4D; see A.3 for how modularity was computed). This toy experiment demonstrates the feasibility of accurately recovering accurate position representations in nonlinear stochastic networks. We next explore this application to a real world brain imaging dataset.

## 5    Experiment 3: Masked prediction of spontaneous brain imaging data

In neuroscience, an important goal is to be able to predict distributed brain activity using the activity of other brain regions (Bassett and Sporns, 2017). To achieve this, we sought to build a generalizable transformer model that would predict the brain activity of target regions using the brain activity of other regions. In this context, we conceptualized 'tokens' as distinct brain regions across the cortical mantle with the goal of predicting masked brain activity. This task can be formalized as masked pretraining, where the input tokens to the model are contemporaneous brain activity across different brain regions with masked (or missing) activity values (Fig 5B). This requires the model to predict missing brain activity from its surrounding context, i.e., the brain activity values of other brain regions.

Naively, the PE of brain regions might manifest as their physical location in 3D space. However, decades of neuroscience research has revealed a modular brain organization, whereby different brain regions belong to distinct functional networks (or communities) (Power et al., 2011; Yeo et al., 2011; Ji et al., 2019; Schaefer et al., 2018). Thus, the goal of using this real-world dataset was to assess the degree to which

models could recover this "modular" network organization in terms of PE. We used publicly available human functional magnetic resonance imaging (fMRI) data from the Human Connectome Project (HCP) dataset (`www.humanconnectomeproject.org`). We used the resting-state fMRI data from a subset ($n = 100$; $n_{train} = 70$; $n_{test} = 30$) of the HCP 1200 participant pool (Van Essen et al., 2013). fMRI data were partitioned into 360 distinct cortical regions (tokens), which mapped onto to 12 distinct functional networks (Ji et al., 2019; Glasser et al., 2016). Additional details of the fMRI data and preprocessing pipelines can be found in Appendix A.2.

### 5.1 Learning interpretable PEs in human brain data for generalized brain activity prediction

We trained transformer models with a mix of fixed and learnable PEs (fixed: `1d-fixed`, `1d-relative`, `1d-rope`, `random`; learnable PEs initialized with $\sigma \in \{0.1, 0.2, 1.0, 2.0\}$), minimizing the MSE of masked brain activity. Figure 5 shows results of the training with 50% masking (Fig. 5C), and testing with 90% masking (Fig. 5D). (We show results with 15%, 75%, and 90% mask training in Appendix Figs. A12,A13,A14) Successful generalization of the trained model involved predicting the masked brain activity of *a separate subject's data* (i.e., test subjects). As expected, we found that small-norm initialized PEs (`learn-0.1` and `learn-0.2`) achieved the best performance in both their training and validation sets after a fixed number of training steps (50k). Interestingly, relative PEs fared the worst, followed by absolute `1d-fixed` and `random` PEs.

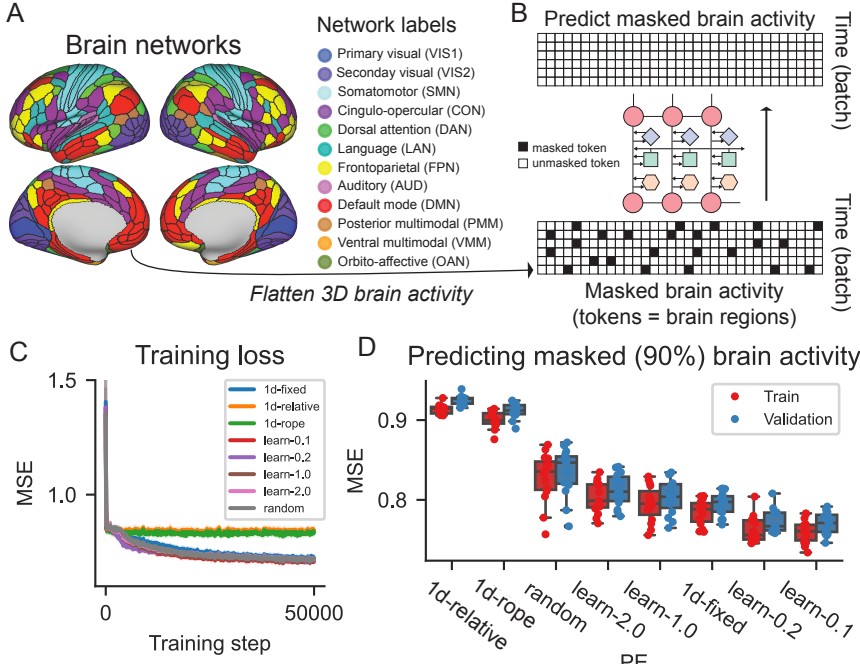

Figure 5: A) The brain is organized into functional networks that are spatially distributed throughout the brain in 3D space. B) We sought to understand the influence of PE on a model's ability to predict contemporaneous brain activity. This can be formalized as a masked prediction task, where a transformer is trained to predict contemporaneous, spatially masked brain activity. To test for generalization, we evaluated the MSE on a validation dataset, which involved predicting the masked activity of data collected from different participants with increased masking (90%). (Models were trained with a 50% masking. See Figures A12,A13,A14 for results with 15%, 75%, and 90% masked training objectives.) C) We found that transformers endowed with different PE schemes converged to different MSE loss values. D) Notably, learnable PEs initialized with small norms (`learn-0.1` and `learn-0.2`) achieved to the lowest MSE for both training and validation datasets. (X-axis is sorted from highest to lowest MSE.)

Having established the superior predictive performance of richly learned PEs (small $\sigma$), we next evaluated whether these models learned meaningful yet nontrivial position information from masked pretraining of brain activity. While the PE of brain regions could be encoded as their physical location in 3D space, decades of neuroscience research has revealed a modular brain network organization, whereby different brain regions

that are spatially distributed throughout the cortex belong to distinct functional networks (or communities) (Fig. 5A) (Power et al., 2011; Yeo et al., 2011; Ji et al., 2019; Schaefer et al., 2018). In other words, two brain regions that are distant in 3D space may actually be "functionally" close (e.g., yellow regions in Fig. 5A). (Prior work has indicated this functional closeness is determined by anatomical connectivity; Vázquez-Rodríguez et al. (2019).) We therefore sought to address whether richly learned PEs could recover this modular functional organization by measuring the distance of learned PE parameters.

When flattening brain regions across the cortex into a 1D tensor, each brain region's (i.e., token's) assignments are distributed across that tensor (Fig. 6A). (In this context, a PE scheme that places adjacent tokens closer to each other, such as in the original `1d-fixed` PE with sines and cosines from Vaswani et al. (2017), would be clearly ill-suited.) To evaluate whether learned PEs learned a modular organization that reflected the known functional network organization of the brain, we measured the distance between every pair of tokens' PE. This involved computing an orthogonal Procrustes transform to rotate and match the embedding dimensions of each token's PE prior to computing their distance (L2 norm) (Figure 6B). The reason this is necessary is because since learned PEs are first randomly initialized, the embedding dimension of each token's PE are not necessarily aligned (e.g., position 1's embedding dimension $i$ does not necessarily correspond to embedding dimension $i$ of position 2). After aligning the embedding dimensions across tokens, we computed the distance between every pair of tokens. (We then scaled distance in this 2D matrix to range from 0 and 1, and computed the complement (i.e., $1 - d_{scaled}$), such that closer PEs would have higher values. We computed both the network modularity and network clustering (i.e., segregation) with respect to the known network partitions (see A.3 for mathematical definitions; Rubinov and Sporns (2010)). In brief, modularity is a statistic that quantifies the degree to which the distance matrix can be cleanly subdivided into the brain's network partitions. Network clustering is a statistic that quantifies the ratio between within-module distances and across-module distances, where modules are defined using a network partitioning from Ji et al. (2019). We found that the modularity of small-norm initialized PEs (`learn-0.1` and `learn-0.2`) had the highest overall network modularity and segregation relative to other learnable PEs. This implies that the small-norm initialized PEs learned interpretable PEs with respect to the known functional networks of the brain. These findings support the hypothesis that learnable PEs (as opposed to off-the-shelf PEs) in the rich training regime can improve brain activity prediction, while successfully learning interpretable position information (Figure 6C,D).

## 6 Discussion

**Related work.** Recent studies have revealed that the choice of PE can strongly influence transformer generalization (Li et al., 2024; Kazemnejad et al., 2023; Golovneva et al., 2024; Ruoss et al., 2023; McLeish et al., 2024; Shen et al., 2024; Csordás et al., 2021; Ontanon et al., 2022; Zhou et al., 2024; Zhang et al., 2024). However, most of these investigations have been limited to evaluating the generalizability of various PE schemes on 1D string-based tasks (e.g., sequence learning tasks for arithmetic, context-free grammars, or compositional tasks). In contrast, many important problems require the encoding of sequences that are not in 1D (e.g., Li et al. (2021)), and where position information is non-trivial or not known, which we investigate here. Additionally, work in deep learning theory has provided insight into the impact of model initialization on representation learning, yet focus primarily on simple neural networks rather than transformers (Chizat et al., 2020; Woodworth et al., 2020; Jacot et al., 2020; Kunin et al., 2024; Lippl and Stachenfeld, 2024). In this study, we apply insights from deep learning theory to transformer models to effectively learn (and improve generalization) to nontrivial sequence tasks, such as tasks requiring reasoning in 2D, or tasks in which ground truth position information is organized in higher dimensions.

**Limitations and future directions.** We have demonstrated that learnable PEs initialized from small-norm distributions can 1) approximate the ground truth PE, and 2) outperform many commonly-used PEs. However, there remain several limitations of the present study which future studies can explore. First, though we consider the use of the LST (with a 2D organization) a strength of this study due to the visual interpretability of the paradigm's positional information, it is unclear how well this approach will generalize to tasks with an arbitrary number of elements or tasks in which there are dynamic changes in the number of elements (e.g., length generalization problems in higher dimensions). In addition, due to the task-dependent nature of utilizing (or learning) optimal PEs, for some tasks and training objectives, such as generic next-

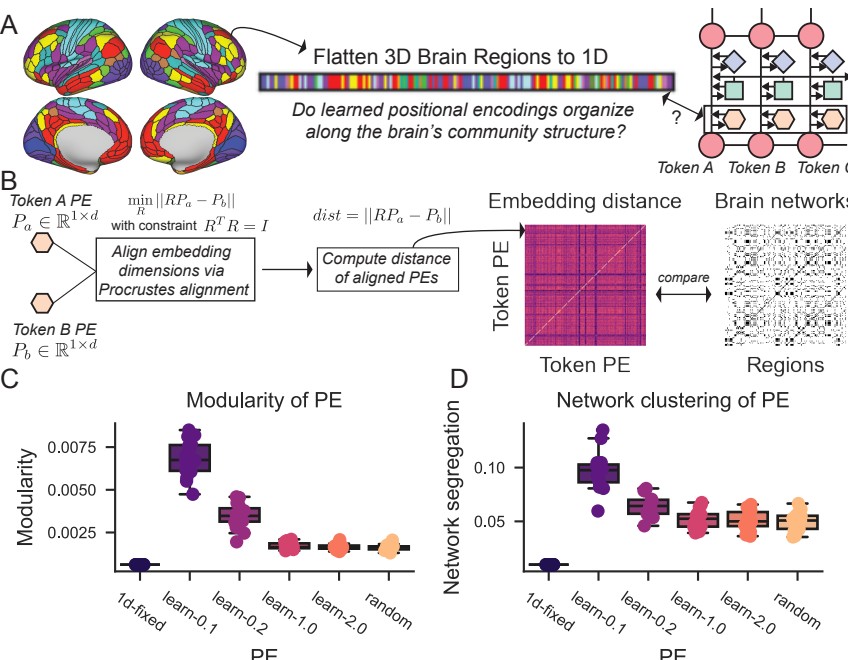

Figure 6: A) We evaluated whether learnable PEs could discover the brain's modular functional network organization. When flattening the brain from a 3D to 1D tensor, the network labeling of each brain region is heterogenously distributed across a 1D tensor in a disorganized manner. B) We computed the distance between positional embeddings between every pair of token positions after aligning their embedding dimensions ($d$) through an orthogonal Procrustes transform. This allowed us to construct a token-by-token embedding distance matrix, which we then compared to brain network organization. C) The modularity of PEs with respect to the brain's network organization. Models that had learnable PEs initialized from small-norm distributions learned a modular PE organization that was consistent with previously identified brain networks (Ji et al., 2019) D) The network clustering of PEs, which assessed whether PEs of tokens that belong to the same network are closer in space than PEs that do not belong to the same network.

token prediction or arithmetic (which is order invariant under addition), standard PE choices may be most appropriate (e.g., `1d-fixed` or `rndpe`). However, for tasks in which establishing an underlying ordering and relation of tokens is crucial — such as reasoning tasks in 2D or tasks with complex network structures, as is common in biology — our results show that using small-norm initialized learnable PEs can be highly beneficial. Second, the current learned PE is limited insofar that the embedding is linearly superimposed on a given token (i.e., $token + pe$). While this makes it potentially difficult to generalize to more complex tasks, a natural future research direction would be to learn nonlinear PE embeddings that allow for PEs to be flexibly generated as a *function* of the token embedding, thereby learning token abstractions/types (i.e., $pe(token)$). This nonlinear formulation of PE as a function of the token embedding would, in theory, have significantly greater expressive ability, and potentially endow transformers with the ability to recognize more complex formal languages (Merrill et al., 2024). Finally, while we have empirically demonstrated that PE initialization impacts the learned representation and generalization in transformers, in practice, it remains unclear how the precise choice of $\sigma$ interacts with other hyperparameters (e.g., architectural choices and optimization protocols). For example, while theory alone suggests small norm initializations should produce rich representations, in practice, such initializations can be hard to train due to small gradients (Chizat et al., 2020). This can be empirically observed by the experiments we performed when using $\sigma=0.01$ and $\sigma=0.05$ (Fig. A4), where we find slower training convergence with smaller $\sigma$ using SGD. Interestingly, however, when Adam is used, $\sigma = 0.01$ leads faster training convergence yet poor generalization, which is likely due to Adam's adaptive learning rates. Specifically, since the learning rate in Adam is scaled by the magnitude of the gradient flow in Adam, this behavior likely produces large changes to the weights (during the initial small gradient updates) that pushes learning regime back into the kernel learning regime. Nevertheless, it will be important for future theoretical work to more rigorously characterize these behaviors to better understand how the choice of $\sigma$, architecture, and optimizer interact to influence representation learning.

**Conclusion**. There are many tasks and problems in which it is difficult to know the ground truth arrangement of input sequences, or tasks in which commonly-used PEs are ill-suited. Examples of such problems include reasoning on parse trees and directed graphs (where node distances and relations are not preserved when flattening into a 1D sequence for transformer sequence processing), or inference on real-world biological datasets in which the ground truth structure is important for prediction yet difficult to know (e.g., the 3D neuroscience data explored here, or co-expression of genes in a DNA sequence based on 3D chromatin conformation) (Szabo et al., 2019; Ji et al., 2021). In this study, we sought to understand how to learn position information directly from data using insights from deep learning theory. In particular, we found that an optimally-learned PE 1) outperformed commonly-used PEs, 2) learned attention maps and PE embeddings that were closely aligned to ground truth position information, and 3) enhanced generalization performance. *Critically, learning an optimal and interpretable PE depended on its initialization in a reasoning task, a nonlinear netowrk simulation, and a real-world biological dataset.* We anticipate these results will spur future investigations into the importance and utility of learnable PEs for structured learning and generalization.

## 7 Ethics statement

All human brain data used in this study is publicly available archived data from the Human Connectome Project (`www.humanconnectomeproject.org`). All participants gave signed, informed consent in accordance with the protocol approved by the local institutional review board.

## 8 Code availability

Code and task environments associated with reproducing models, results, and figures in this study will be made publicly available.

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

# A    Appendix / supplemental material

## A.1    Positional encoding definitions

Below, we provide the formal definitions for the common PEs that we evaluated learnable PEs against: `1d-fixed`, `2d-fixed`, `1d-relative`, and `1d-rope`.

**1d-fixed** (Vaswani et al., 2017). 1D absolute PEs were defined from Vaswani et al. (2017) (Vaswani et al., 2017). For a given position *pos* (here $1 \leq pos \leq 16$, for elements in the LST grid), we define

$$PE_{(pos,2i)} = sin(\frac{pos}{10000^{2i/d_{model}}})$$

$$PE_{(pos,2i+1)} = cos(\frac{pos}{10000^{2i/d_{model}}})$$

where $i$ is the embedding dimension, and $d_{model}$ is the dimensionality of the embedding vector. Note that $PE_{(pos,2i)}$ (sines) is reserved for even embedding dimensions, and $PE_{(pos,2i)}$ (cosines) is reserved for odd embedding dimensions.

**2d-fixed**. 2D absolute PEs were a 2D generalization of `1d-fixed` (Vaswani et al., 2017). The primary distinction is that rather than $1 < pos < 16$, there are two position variables, $1 \leq pos_w \leq 4$ and $1 \leq pos_h \leq 4$ (for width and height of grid). Positional encoding for a row $w$ is defined by

$$PE_{(pos_w,2i)} = sin(\frac{pos_w}{10000^{2i/d_{model}}})$$

$$PE_{(pos_w,2i+1)} = cos(\frac{pos_w}{10000^{2i/d_{model}}})$$

Positional encoding for a column $h$ is defined by

$$PE_{(pos_h,2i)} = sin(\frac{pos_h}{10000^{2i/d_{model}}})$$

$$PE_{(pos_h,2i+1)} = cos(\frac{pos_h}{10000^{2i/d_{model}}})$$

For a 2D PE encoding, half the embedding dimensionality is reserved for encoding rows; the other half of the embedding dimensionality is reserved for encoding columns. Thus, for $d_{model} = 160$, embedding dimensions 0-79 are reserved for encoding rows. Embedding dimensions 80-159 are reserved for encoding columns.

**1d-relative** (Shaw et al., 2018). `1d-relative` PE modifies standard self-attention to incorporate the relative positions of tokens. This implies that calculation of PE is wrapped within the self-attention module. The relative position embedding parameter between a token at position $i$ and $j$ is $a_{j-1}$. In brief, self attention is then modified to include relative position information by modifying attention between tokens $i$ and $j$ as

$$e_{ij} = \frac{x_i W^Q (x_j W^K)^T + x_i W^Q (a_{ij}^K)^T}{\sqrt{d_z}}$$

where $x_i$ and $x_j$ are the embeddings for tokens $i$ and $j$, and $W^Q$ andd $W^K$ are the query and key matrices, respectively. Additional details can be found in (Shaw et al., 2018).

**1d-rope** (Su et al., 2022). `1d-rope` applies a rotation to the token embeddings based on their positions in a higher dimensional space. For a token at position $p$ with an embedding $x$, let $x = [x_1, x_2, ..., x_d]$, where $d$ is even. Then, for each pair of dimensions, apply the rotation

$$\begin{pmatrix} \hat{x}_{2k} \\ \hat{x}_{2k+1} \end{pmatrix} = \begin{pmatrix} cos(\theta_p) & -sin(\theta_p) \\ sin(\theta_p) & cos(\theta_p) \end{pmatrix} \begin{pmatrix} x_{2k} \\ x_{2k+1} \end{pmatrix}$$

with $\theta_p = \frac{p}{10000^{2k/d}}$. The resulting embedding is the concatenation of rotated pairs. Additional details can be found in the original paper (Su et al., 2022).

## A.2  fMRI data and preprocessing

Data were previously collected as part of the Human Connectome Project and made publicly available (Van Essen et al., 2013). All participants gave signed, informed consent in accordance with the protocol approved by the Washington University institutional review board. Whole-brain multiband echo-planar imaging acquisitions were collected on a 32-channel head coil on a modified 3T Siemens Skyra with TR = 720 ms, TE = 33.1 ms, flip angle = 52°, Bandwidth = 2,290 Hz/Px, in-plane FOV = 208x180 mm, 72 slices, 2.0 mm isotropic voxels, with a multiband acceleration factor of 8. Data were collected across two days, with the first two resting-state fMRI sessions collected on the first day, and another two sessions collected on the second day. Each resting-state fMRI session lasted 14.4 minutes. Additional details on imaging sessions can be found in Smith et al. (2013).

Data were provided in a minimally preprocessed format. We performed additional preprocessing steps in accordance with Ito et al. (2020), which we paraphrase below. We first parcellated minimally preprocessed data into 360 brain regions using the Glasser et al. (2016). In addition, we removed the first five frames of each run, de-meaning and de-trending the timeseries, and performing nuisance regression on the minimally preprocessed data. Nuisance regression included removing motion signals and physiological noise. Six primary motion parameters were included, along with their derivatives and quadratic timeseries. Physiological noise was modeled using aCompCor on the timeseries extracted from the white matter and ventricles (Behzadi et al., 2007). For aCompCor, the first 5 principal components from the white matter and ventricles were extracted separately and included in the nuisance regression. We also included the derivatives of each of those components, and the quadratics of all noise regressors. In total, the nuisance regression modeled contained 64 nuisance regressors.

## A.3  Modularity and clustering in network data

Modularity and network clustering (segregation) are common measures in network science, and commonly applied to fMRI data (Rubinov and Sporns, 2010). We adopted these measures to calculate the modularity and network clustering of learned PE parameters. The distance $d$ between PEs was scaled between 0 and 1, and we calculated the complement $(1 - d)$ such that higher values indicated two PEs were closer. Both modularity and network clustering were calculated with respect to the predefined network partition. Modularity $Q^W$ of the learned, PE distance matrix $W$ was calculated as

$$Q^W = \frac{1}{l^W} \sum_{i,j \in N} \left[ W_{ij} - \frac{k_i^W k_j^W}{l^W} \right] \delta_{m_i, m_j}$$

where $l^W$ is the sum of all weights in $W$, $N$ are tokens (brain regions), $W_{ij}$ is the distance between token $i$ and token $j$, $k_i^W$ is the weighted degree of token $i$, $m_i$ is the module containing node $i$, and $\delta_{m_i, m_j} = 1$ if $m_i = m_j$ and 0 otherwise (as determined by the network partition).

Network clustering $C$ is measured as the difference in mean within module and across module distances, as a proportion of the within-module distance

$$C = \frac{1}{|M|} \sum_{m \in M} \left[ \frac{\bar{W}_{m_{in}} - \bar{W}_{m_{out}}}{\bar{W}_{m_{in}}} \right]$$

where $M$ is the full set of modules (networks), $\bar{W}_{m_{in}}$ is the mean within-module distance, and $\bar{W}_{m_{out}}$ is the across-module distance.

Table A2: Training and generalization performance for all learnable PE models initialized with different $\mathcal{N}(0,\sigma)$ (no regularization). This is the corresponding data table for Fig. 2.

| PE | Training acc | Training SD | Validation acc | Validation SD |
|---|---|---|---|---|
| 0.100 | 1.000 | 0.000 | 0.925 | 0.083 |
| **0.200** | **1.000** | **0.000** | **0.956** | **0.039** |
| 0.300 | 1.000 | 0.000 | 0.946 | 0.022 |
| 0.400 | 1.000 | 0.000 | 0.929 | 0.048 |
| 0.500 | 1.000 | 0.000 | 0.946 | 0.025 |
| 0.600 | 1.000 | 0.000 | 0.929 | 0.027 |
| 0.700 | 1.000 | 0.000 | 0.926 | 0.027 |
| 0.800 | 1.000 | 0.000 | 0.916 | 0.048 |
| 0.900 | 1.000 | 0.000 | 0.896 | 0.060 |
| 1.000 | 1.000 | 0.000 | 0.894 | 0.043 |
| 1.100 | 1.000 | 0.000 | 0.835 | 0.101 |
| 1.200 | 1.000 | 0.000 | 0.756 | 0.141 |
| 1.300 | 1.000 | 0.000 | 0.702 | 0.126 |
| 1.400 | 1.000 | 0.000 | 0.654 | 0.161 |
| 1.500 | 1.000 | 0.000 | 0.616 | 0.208 |
| 1.600 | 1.000 | 0.000 | 0.582 | 0.168 |
| 1.700 | 1.000 | 0.000 | 0.480 | 0.167 |
| 1.800 | 1.000 | 0.000 | 0.378 | 0.153 |
| 1.900 | 1.000 | 0.000 | 0.406 | 0.161 |
| 2.000 | 1.000 | 0.000 | 0.377 | 0.171 |

Table A3: Attention map similarity between learnable PEs and the ground truth `2d-fixed`. Corresponding data table for Fig. 3.

| $\sigma$ | Cosine | SD |
|---|---|---|
| 0.100 | 0.836 | 0.018 |
| 0.200 | 0.838 | 0.016 |
| 0.300 | 0.808 | 0.013 |
| 0.400 | 0.779 | 0.013 |
| 0.500 | 0.748 | 0.014 |
| 0.600 | 0.722 | 0.016 |
| 0.700 | 0.715 | 0.016 |
| 0.800 | 0.710 | 0.016 |
| 0.900 | 0.694 | 0.015 |
| 1.000 | 0.685 | 0.015 |
| 1.100 | 0.697 | 0.015 |
| 1.200 | 0.692 | 0.017 |
| 1.300 | 0.685 | 0.016 |
| 1.400 | 0.672 | 0.016 |
| 1.500 | 0.657 | 0.017 |
| 1.600 | 0.647 | 0.022 |
| 1.700 | 0.642 | 0.016 |
| 1.800 | 0.624 | 0.021 |
| 1.900 | 0.606 | 0.023 |
| 2.000 | 0.577 | 0.019 |

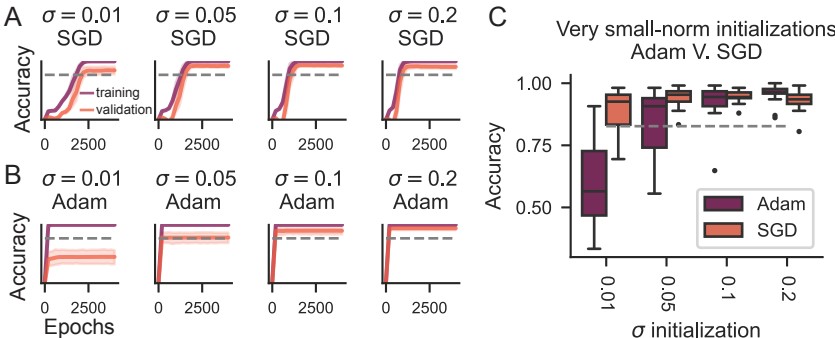

Figure A7: Different optimizers (Adam, SGD) can lead to different generalization performances in very small-norm PE initializations. We noticed that for models that were initialized with very small-norm PEs, generalization performance was reduced, counter to the theoretical claims in the NTK theory. However, we realized that this could be attributed to variability in the adaptive learning rates inherent to the Adam optimizer. Thus, we evaluated whether using vanilla SGD (learning rate=0.001) would ameliorate these reduced accuracies. Indeed, we found that for very small-norm initializations, SGD tended to ameliorate the reduced generalization effects observed with Adam. A) Training trajectories for $\sigma \in 0.01, 0.05, 0.1, 0.2$ using vanilla SGD (learning rate=0.001). Note that the slow training is significantly more obvious for small-norm initializations with SGD, consistent with theory. B) Corresponding training trajectories using Adam. C) A clear generalization discrepancy emerges when using Adam vs. SGD.

Table A4: Performance differences across different optimizers (Adam, SGD) in very small-norm PE initializations (without regularization). Corresponding data table for Fig. A7C.

| $\sigma$ | Adam acc | Adam sd | SGD acc | SGD sd |
|---|---|---|---|---|
| 0.010 | 0.588 | 0.186 | 0.885 | 0.101 |
| 0.050 | 0.836 | 0.140 | 0.941 | 0.042 |
| 0.100 | 0.925 | 0.083 | 0.947 | 0.027 |
| 0.200 | 0.956 | 0.039 | 0.930 | 0.044 |

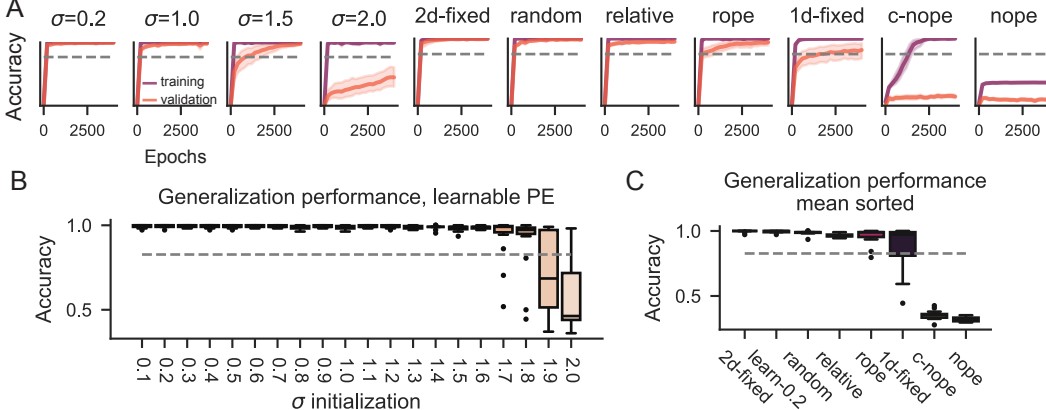

Figure A8: Generalization performance of learnable and common PEs with weight decay (0.1). To assess the impact of learnable PEs in a more realistic setting, we measured the effect of L2 regularization with a weight decay parameter of 0.1. In general, weight decay improves the generalization of learnable PE models across initializations, except for particularly high $\sigma$. Importantly, learnable PEs with weight decay become nearly indistinguishable to the ground-truth PE. Nevertheless, the general pattern remains for learnable PE models: the smaller the $\sigma$, the greater the generalization. A) Training and validation trajectories across training. B) Generalization performance for learnable PE models with weight decay = 0.1. C) Generalization performance for common PE models with weight decay = 0.1.

Table A5: Training and validation performance of learnable PEs with weight decay (0.1). Corresponding data table for Fig. A8B.

| $\sigma$ | Validation acc | Validation SD | Training acc | Training SD |
|---|---|---|---|---|
| 0.100 | 0.996 | 0.006 | 1.000 | 0.000 |
| 0.200 | 0.995 | 0.007 | 1.000 | 0.000 |
| 0.300 | 0.994 | 0.008 | 1.000 | 0.000 |
| 0.400 | 0.994 | 0.007 | 1.000 | 0.001 |
| 0.500 | 0.993 | 0.007 | 0.999 | 0.002 |
| 0.600 | 0.993 | 0.007 | 1.000 | 0.001 |
| 0.700 | 0.993 | 0.009 | 0.999 | 0.003 |
| 0.800 | 0.993 | 0.009 | 0.999 | 0.003 |
| 0.900 | 0.993 | 0.008 | 1.000 | 0.001 |
| 1.000 | 0.993 | 0.008 | 1.000 | 0.001 |
| 1.100 | 0.990 | 0.008 | 0.999 | 0.002 |
| 1.200 | 0.989 | 0.010 | 1.000 | 0.001 |
| 1.300 | 0.989 | 0.012 | 1.000 | 0.000 |
| 1.400 | 0.987 | 0.010 | 0.999 | 0.002 |
| 1.500 | 0.985 | 0.013 | 0.999 | 0.002 |
| 1.600 | 0.983 | 0.018 | 0.999 | 0.002 |
| 1.700 | 0.929 | 0.138 | 1.000 | 0.001 |
| 1.800 | 0.900 | 0.180 | 1.000 | 0.001 |
| 1.900 | 0.707 | 0.238 | 0.992 | 0.024 |
| 2.000 | 0.586 | 0.238 | 0.996 | 0.012 |

Table A6: Training and validation performance of common PEs with weight decay (0.1). Corresponding data table for Fig. A8C.

| PE | Validation acc | Validation SD | Training acc | Training SD |
|---|---|---|---|---|
| 2d-fixed | 0.997 | 0.008 | 0.999 | 0.001 |
| learn-0.2 | 0.994 | 0.008 | 1.000 | 0.000 |
| random | 0.987 | 0.016 | 0.999 | 0.003 |
| 1d-relative | 0.965 | 0.012 | 0.999 | 0.004 |
| 1d-rope | 0.959 | 0.060 | 1.000 | 0.001 |
| 1d-fixed | 0.872 | 0.166 | 0.998 | 0.001 |
| c-nope | 0.352 | 0.036 | 0.999 | 0.003 |
| nope | 0.322 | 0.017 | 0.508 | 0.002 |

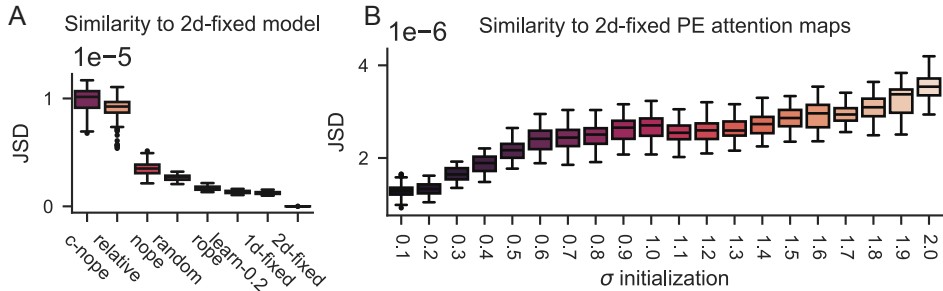

Figure A9: We computed the Jensen-Shannon Divergence of the attention weights for every learnable PE model and the ground truth `2d-fixed` model. This Figure is a comparable analysis to Fig. 3, but using the Jensen-Shannon Divergence distance metric applied to attention maps (instead of cosine similarity).

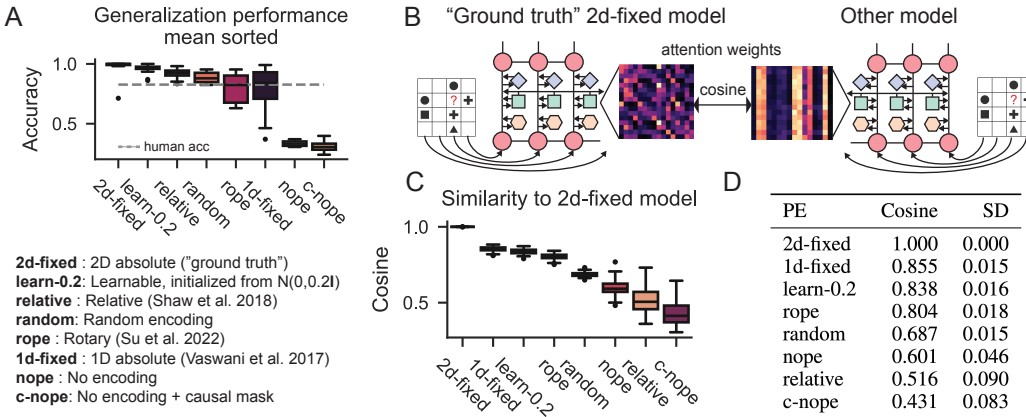

Figure A10: Comparing top-performing learnable PE models (`learn-0.2`) with commonly-used PE schemes. A) We compare the generalization performance of `learn-0.2` model with models using common PEs. Notably, we included a `2d-fixed` PE as the "ground truth", since it obeyed the 2D organization of the LST task. The `learn-0.2` model outperformed all other PE models. B) We compared the attention maps of each model to those derived from the "ground truth" `2d-fixed` model. C,D) We found that aside from the `1d-fixed` model (which does not generalize well), `learn-0.2` learned the closest attention map to the `2d-fixed` model. This was expected, since the `2d-fixed` and `1d-fixed` PEs are highly similar by design (the baseline cosine similarity between the two schemes is 0.72). In contrast, the `learn-0.2` learned an attention map that was highly similar to the `2d-fixed` model, despite having no similarity to the `2d-fixed` PE scheme at initialization (cosine at initialization = 0.00). (We also note that the `1d-rope` model had high baseline similarity to the `2d-fixed`, since by construction, a component of the `1d-rope` encoding is highly similar to the `1d-fixed` PE scheme.) Thus, we found that despite having no prior bias towards the `2d-fixed` PE scheme, a small-norm initialized learnable PE is capable of learning an attention map that approximates an attention map derived from the ground truth PE.

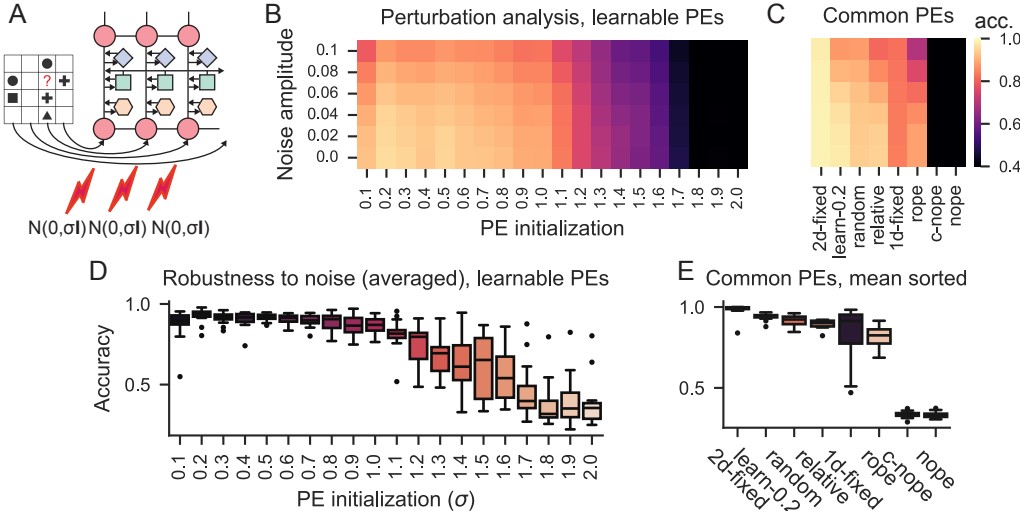

Figure A11: A) We injected noise with different amplitudes into each token embedding, and assessed downstream generalization performance. B) Generalization performance with noisy inputs across all initializations for learnable PEs, and C) common PEs. D) Average performance across noisy inputs (collapsing across rows in B). E) Besides the ground truth model (`2d-fixed`), the `learn-0.2` model was most robust to perturbation compared to common PEs.

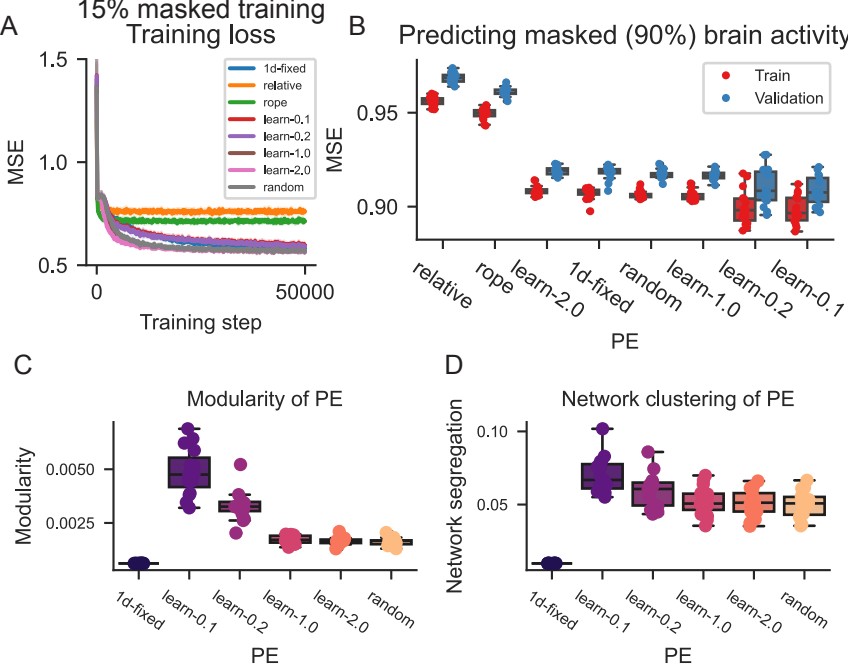

Figure A12: Training and evaluating models on fMRI data with 15% masked pretraining. A) Training trajectory for each model with 15% masked pretraining. B) MSE of each model on training and testing datasets at the end of training (after 50k training steps) predicting on 90% masked input. (X-axis is sorted by highest to lowest MSE.) Consistent with results in the main text, models with a learnable PE parameter (initialized from a small-norm distribution) achieved the lowest generalization MSE. C) The modularity of PEs with respect to the brain's network organization (analogous analysis to Fig. 6C). D) The network clustering of PEs within a model, which assessed whether PEs of tokens that belong to the same network are closer in space than PEs that do not belong to the same network (analogous analysis to Fig. 6D). Consistent with results in the main text, models with a learnable PE parameter initialized from a small-norm distribution learned a similar network modularity/clustering consistent with the brain's known network organization.

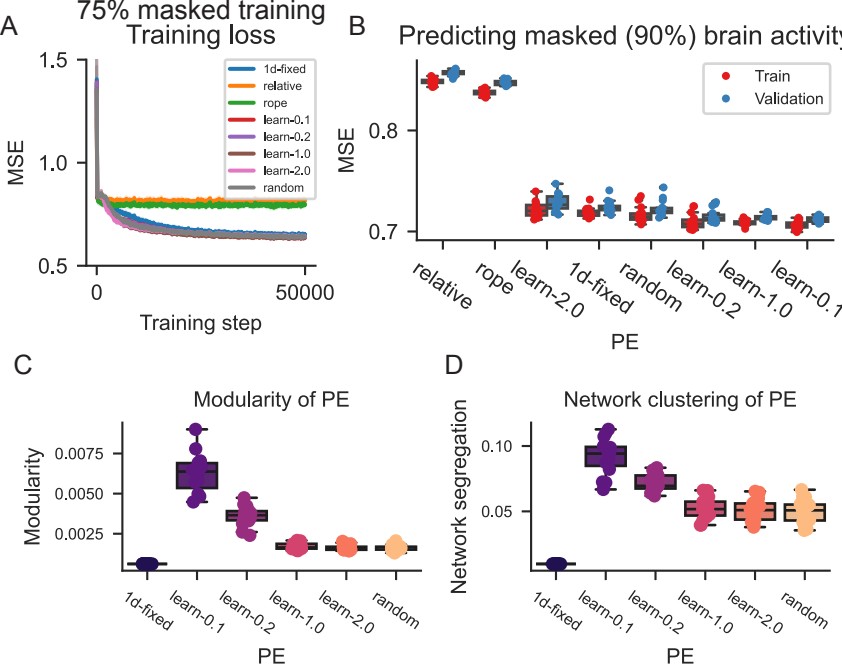

Figure A13: Training and evaluating models on fMRI data with 75% masked pretraining. A) Training trajectory for each model with 75% masked pretraining. B) MSE of each model on training and testing datasets at the end of training (after 50k training steps) predicting on 90% masked input. (X-axis is sorted by highest to lowest MSE.) Consistent with results in the main text, models with a learnable PE parameter (initialized from a small-norm distribution) achieved the lowest generalization MSE. C) The modularity of PEs with respect to the brain's network organization (analogous analysis to Fig. 6C). D) The network clustering of PEs within a model, which assessed whether PEs of tokens that belong to the same network are closer in space than PEs that do not belong to the same network (analogous analysis to Fig. 6D). Consistent with results in the main text, models with a learnable PE parameter initialized from a small-norm distribution learned a similar network modularity/clustering consistent with the brain's known network organization.

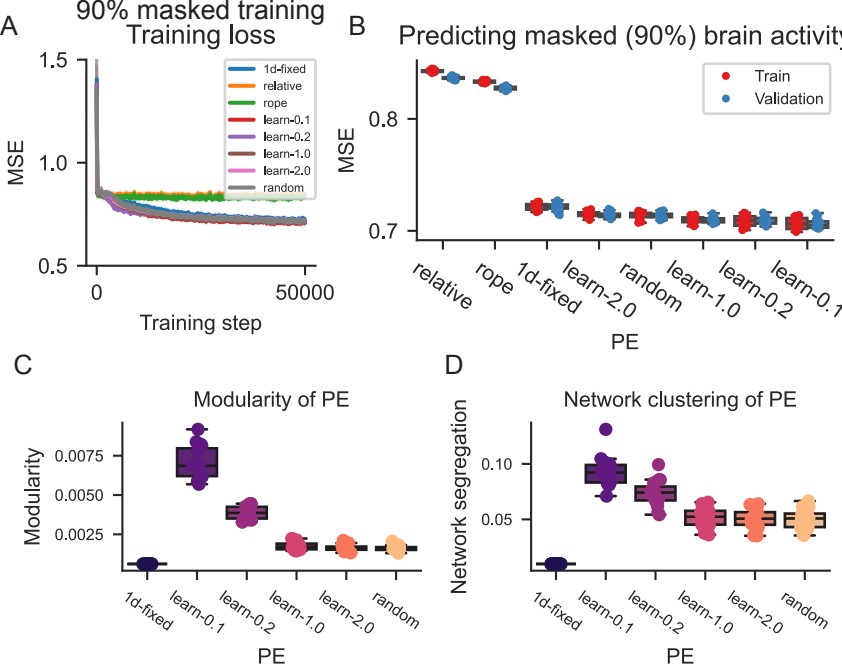

Figure A14: Training and evaluating models on fMRI data with 90% masked pretraining. A) Training trajectory for each model with 90% masked pretraining. B) MSE of each model on training and testing datasets at the end of training (after 50k training steps) predicting on 90% masked input. (X-axis is sorted by highest to lowest MSE.) Consistent with results in the main text, models with a learnable PE parameter (initialized from a small-norm distribution) achieved the lowest generalization MSE. C) The modularity of PEs with respect to the brain's network organization (analogous analysis to Fig. 6C). D) The network clustering of PEs within a model, which assessed whether PEs of tokens that belong to the same network are closer in space than PEs that do not belong to the same network (analogous analysis to Fig. 6D). Consistent with results in the main text, models with a learnable PE parameter initialized from a small-norm distribution learned a similar network modularity/clustering consistent with the brain's known network organization.

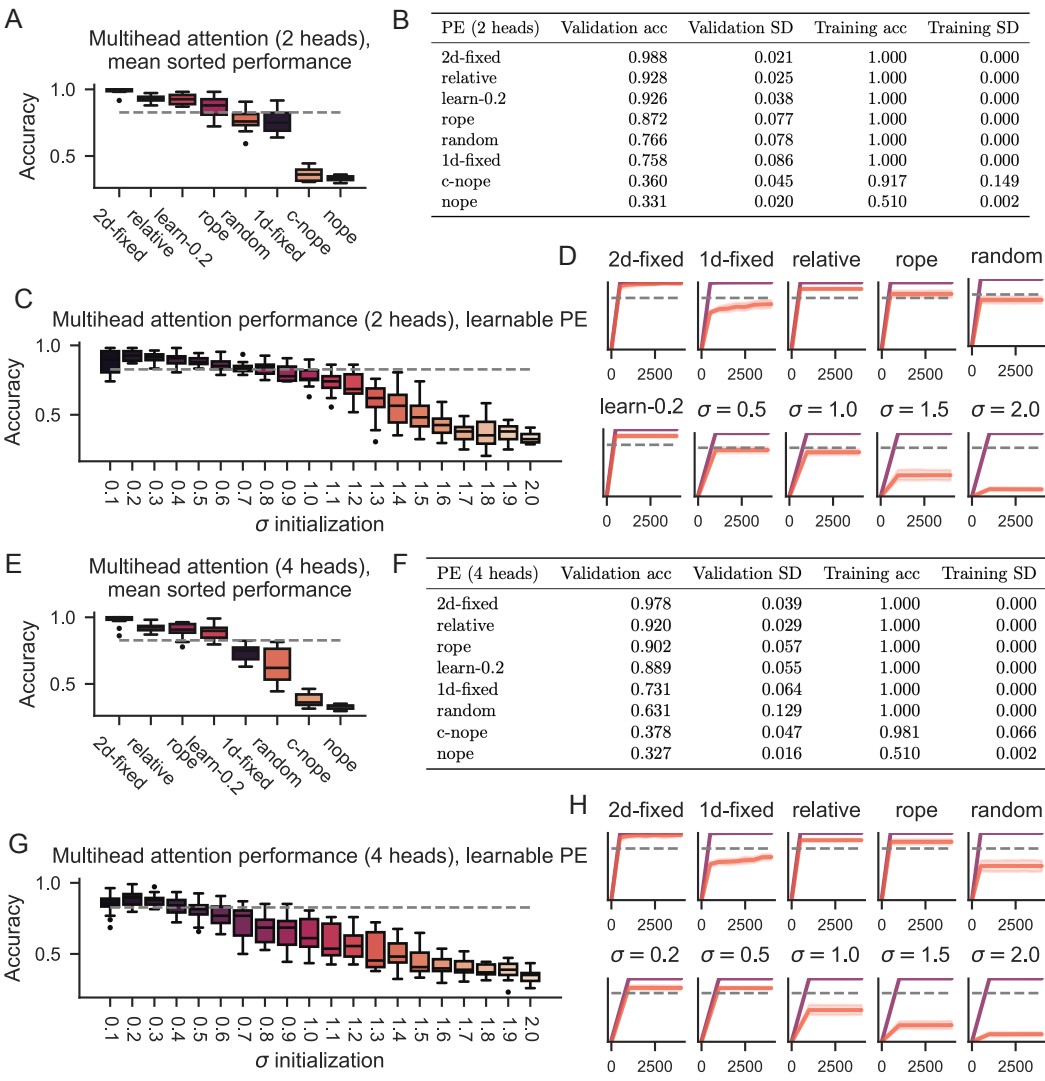

Figure A15: Model performances when incorporating multihead (2 and 4) attention mechanisms. Overall, we find that adding attention heads tends to reduce generalization performance across the board. The only architecturein which we see improvements are models with `1d-rope` PE. Nevertheless, despite their improvements, `1d-rope` models still do not outperform models with `1d-relative` PEs, nor do they outperform `learn-0.2` models with a single attention head (single head `learn-0.2` performance = 95.6%; Table 1). We also notably see that `learn-0.2` models tend to degrade in their performance, likely due to the increase in free parameters. A,B) Performance of models with 2 attention heads. C) Performance and of models with learnable PEs (2 attention heads). D) Training trajectories for example model architectures (2 attention heads). E,F) Performance of models with 4 attention heads. G) Performance of models with learnable PEs (4 attention heads). H) Training trajectories for example model architectures (4 attention heads).

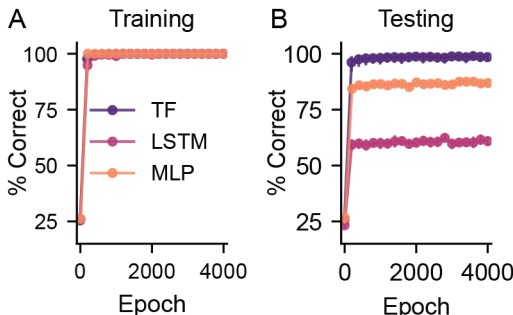

Figure A16: Control models. We assessed two alternative artificial neural network architectures: A vanilla multilayer perceptron (MLP), and a bidirectional Long Short-Term Memory (LSTM) model. Each model contained four layers and 160 hidden units each. Overall, we found that compared to the `2d-fixed` PE transformer model, the MLP and LSTM model could not generalize the LST task.

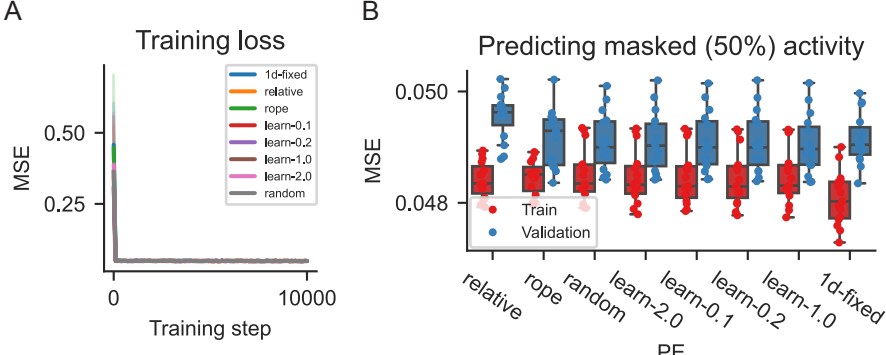

Figure A17: Supplementary data associated with Fig. 4. We report the training and validation MSE evaluation at the end of training for the stochastic network simulation. A) Training converged rapidly across all models. B) Across all the models / PE variants, there was little difference in MSE across training and validation distributions for this task. This is because all models optimally converged. In addition, all models could not converge to near zero due to the stochastic nature of the simulation (which cannot be perfectly modeled).

