# OpenReview forum: "Learning positional encodings in transformers depends on initialization"
_TMLR — Rejected by TMLR_

### Review · Reviewer_PSM3 · 2025-02-15

**Summary Of Contributions:**

This paper systematically investigated how common positional encoding (PE) schemes used in transformers facilitate generalization performance and recover ground-truth token order/structure. Across both synthetic and real-world tasks, the authors show that transformers trained with learnable PE with small-norm initialization consistently achieve better generalization and are more capable of recovering the underlying position structure.

**Audience:**

Yes

**Broader Impact Concerns:**

No major concerns identified.

**Claims And Evidence:**

Yes

**Requested Changes:**

I think the paper is in good shape, but consider the following:
- Clarify the questions raised above.
- I didn't see an accuracy profile across different PE scheme for the simulated network task. It would be good to include that to help interpret Figures 4D and 4F, or give a pointer to it in appendix if it's presented there.
- Fix some minor typos.

**Strengths And Weaknesses:**

Strengths:
- I find the paper a pleasure to read -- good writing and clear presentation overall.
- The results were consistent across a range of interesting synthetic and real-world tasks and link to prior work on lazy vs. rich learning.
- The results furthers our understanding of the role of PEs in transformer task learning and generalization.

Weakness:
- All tasks are bidirectional with a masked objective, so it's unclear whether the results would generalize to models trained with an autoregressive objective or other types of structured attention. There is brief mentioning of this in the Discussion, but I wonder if the authors have further thoughts on how these results may/may not generalize beyond the masked/full-attention objective.

Questions/Clarifications:
- The 1D-fixed PE was able to achieve good accuracy in predicting brain activities but has very low modularity score. Do you have a sense of what it has learned instead? Is the interpretation that there are other ways to reconstruct brain activity that does not rely on the ground-truth modular structure?
- The motivation to have all models trained for the same # of training steps is unclear to me.

---

> ### Author Response · Authors · 2025-03-07
> **Author response to Reviewer PSM3 (1/2)**
>
> We thank the Reviewer for taking the time to assess our manuscript, and for their overall positive assessment of our paper.
>
> **Response to Weaknesses:**
>
> `1. Limitation of tasks using bidirectional attention and masked training objectives`
>
> Indeed, though most of our models used bidirectional attention, we would like to qualify that not all of them used a masked (i.e., self-supervised) objective; the Latin Squares Task paradigm was a classification problem that requires reasoning over the 2d grid to place a new element in the red “?” such that no two symbols are repeated in either a row or a column.
>
> Nevertheless, as alluded to by the Reviewer, none of the tested experiments required an autoregressive objective. However, we did include a single model with a causal mask as a comparison to a recent paper (Kazemnejad et al., 2023), which showed that a causal mask with no explicit positional encoding improved a specific type of generalization (length generalization) for 1D string-based tasks. In practice, this choice of positional encoding generalized poorly on the tasks we tested precisely because of its lack of bidirectional attention (see “c-nope” result in Table 1.). Thus, informed by the present experiments, our broader hypothesis and speculation for autoregressive objectives is the following: Tasks that require inference over multiple dimensions may benefit first from encoding the problem using bidirectional attention (with learned or predesigned positional encodings), which can then be fed to a downstream decoder model that uses the encoder-produced embeddings for autoregressive generation. In fact, this architecture is the same as the original encoder-decoder architecture introduced in (Vaswani et al., 2017). We think it will be valuable for future work to assess this distinction, which will require a more careful and nuanced experimental design that evaluates autoregressive objectives using inputs organized in $n$-dimensions.
>
> **Response to Questions/Clarifications:**
>
> `2. Regarding 1d-fixed PE in predicting masked brain activity`
>
> The 1d-fixed PE is a fixed parameter / encoding, and therefore any ‘learning’ is designated to the attention mechanism. Its low modularity is therefore a simple demonstration of how ill-suited positional encodings (which are commonly used in the literature) are not generically suitable for all problems (e.g., in this case a masked prediction objective for timeseries in 3D). However, as noted by the Reviewer, the 1d-fixed PE model can approximate a generalizable solution that circumvents the poor choice of PE, though not to the extent of a learnable PE initialized from a low-norm distribution (Fig. 5d). This is likely because despite low modularity, when flattening brain regions from 3D to 1D, there is preservation of distance/adjacency preserved in 3D (e.g., of the 360 regions, the first 180 are in the left hemisphere, and the latter 180 are on the right hemisphere). This means that 1d-fixed provides some autocorrelation between adjacent tokens/brain regions that may be useful for prediction, yet do not provide the flexible ability to learn the modular organization of positional encodings that is known to exist in the brain.
>
> `3. Motivation to have all models trained for the same # of training steps`
>
> The alternative we considered was to train models until they converged to some loss or accuracy threshold. However, not all models converged within a reasonable amount of time. For example, the model with the causal attention mask with no explicit positional encoding (c-nope), which has been previously shown to be optimal for 1D string-based length generalization tasks (Kazemnejad et al., 2023), obtained 55.9% training accuracy on the Latin Squares paradigm. Moreover, prior studies have demonstrated that simple feedforward networks initialized from a small-distribution train very slowly due to small gradients, with some providing unrealistic training times (Flesch et al., 2022). Thus, we wanted to ensure that in practice, small initializations would not produce unrealistic training times when learning positional encoding parameters in transformers.

---

> ### Author Response · Authors · 2025-03-07
> **Author response to Reviewer PSM3 (2/2)**
>
> **Other requested changes**
>
> `4. Accuracy profiles across different PE schemes for the simulated network task.`
>
> Thank you for spotting this missing analysis. We have now included the mean squared error (MSE) on the simulated network task (Appendix Fig. A17). Across all the models / PE variants, there was little difference in MSE across testing distributions for this task. This is because all models optimally converged. In addition, all models could not converge to zero due to the stochastic nature of the simulation (which cannot be perfectly modeled due to noise). Despite similar MSE evaluation on the testing distribution, small-norm initialized PEs were the only models that were capable of correctly recovering the ground truth network positions. Overall, this suggests that good generalization does not necessarily imply that the correct representations were learned. This figure has been referenced within the caption of Fig 4 in the main text (p. 8):
>
> “(For additional data on training and testing convergence, see Fig. A17.)”
>
> In addition, the caption to Fig. A17 reads as the following (p. 27):
>
> “Figure A17: Supplementary data associated with Fig. 4. We report the training and validation MSE evaluation at the end of training for the stochastic network simulation. A) Training converged rapidly across all models. B) Across all the models / PE variants, there was little difference in MSE across training and validation distributions for this task. This is because all models optimally converged. In addition, all models could not converge to near zero due to the stochastic nature of the simulation (which cannot be perfectly modeled)”
>
> `5. Minor typos`
>
> Thank you -- we have reviewed the manuscript to fix typos.
>
> Thank you for taking the time to assess our manuscript!

---

### Review · Reviewer_jYDr · 2025-02-19

**Summary Of Contributions:**

This work studies the effect of varying standard deviation, sigma, to initialize the learnable positional encodings (PEs) for transformers, with a focus on the 2D positional encodings. Small values for sigma (\~0.2) are reported to perform better than large values (~2.0) on three tasks, including the Latin Square Task (a toy Sudoku), "stochastic network simulation" (synthetic time series), and masked prediction of brain imaging data (fMRI). Analyses show that learned PEs capture certain properties of the original data/problem.

**Audience:**

Yes

**Claims And Evidence:**

No

**Requested Changes:**

Please see the Weaknesses section above. In my view, light text modifications or a few extra experiments would not be enough to address my concerns.

* Notations given to different positional encoding schemes are confusing, e.g., “relative” and “rope” are “1d-relative” and “1d-rope” in reality. Ideally, their 2D versions should be tested and compared to the learned absolute PE.

* Justify why ~0.1 is the right order of magnitude and not 0.01 or 0.001 to strengthen the relevance of discussing any theory here.

* Include experiments with values for sigma that are smaller than 0.1.

* Replace the toy experiment of Sec 4 by a more conventional dataset (e.g., ViT for image classification) to confirm the claims in a more conventional 2D setting.

* Include results on the test set in all the experiments (not only just training and validation).

If the authors wish to keep the emphasis on generalization:
* Include experiments that actually test for more challenging “generalization” such as length generalization in 2D; e.g., LST with different grid sizes or fMRI with different brain sizes (for different individuals).

* > “In neuroscience, a central goal is to be able to predict distributed brain activity using the
activity of other brain regions”

This sounds like an extremely restricted view on the goals of neuroscience…

* Both Sec 4 and 5 only have one subsection, 4.1. and 5.1; I would recommend using subsections only when there are more than one.

Typo:
Paragraph before Sec 4: “we measured teh alignment”

**Strengths And Weaknesses:**

### **Strengths** ###

2D positional encodings have been studied less than their 1D counterparts, presenting opportunities for interesting research.

Use of brain data (fMRI) for such a study is rather original.

### **Weaknesses** ###

I am not convinced by the soundness of the experimental settings and the resulting conclusions, and therefore, the overall quality of the paper.

It is claimed in the conclusion that:

> In this study, we sought to understand how to learn position information directly from data using insights from deep learning theory. In particular, we found that an optimally-learned PE 1) outperformed commonly-used PEs, 2) learned attention maps and PE embeddings that were closely aligned to ground truth position information, and 3) enhanced generalization performance.

I have concerns with each of these claims.

Regarding:
> understand how to learn position information directly from data using insights from deep learning theory.

I find the overall connection to the NTK theory very weak. The main takeaway from each experiment is essentially: “small-norm initialization outperforms the large-norm counterpart. However, unless there is a theory-based prediction on what it means to be “small” (e.g., as a function of the dimension of the embedding dimension *d*, e.g., *1/d* or *1/sqrt(d)*), I cannot see any broader implications of the results presented here beyond those of a straightforward ablation study (e.g., Table A2). Many ambiguities remain. In particular, the range of “sigma” used in experiments is from 0.1 to 2. What tells us that 0.1 is “small”? Why not 0.01 or 0.001? Would such a definition of “small” change if we used a 1000 times larger embedding size? etc… It is unclear to me what useful “insights from deep learning theory” are highlighted and tested here.

> 1) outperformed commonly-used PEs

Given that the comparison only includes “commonly-used” **1D** PE baselines, this statement is not a strong result. The only 2D PE baseline is “2d-fixed” (which outperforms “learn-0.2” in Table 1). It would make more sense to use the 2D versions of “relative” and “rope” PEs instead of the 1D version tested here.

> 2) learned attention maps and PE embeddings that were closely aligned to ground truth position information,

It is misleading to call the 2D sinusoidal PE (“2d-fixed”) “ground truth” which can be interpreted as being the best way to encode positions, because (i) In the 1D case, there are relative PE variants that outperform 1D sinusoidal PE, so there is no reason to believe that it won’t be the case for the 2D case, and (ii) learned PEs may learn a better way to encode positions than the fixed PE; in this sense, there is no “ground truth” representations in principle. For certain tasks, it may be the case that the learned PEs indeed learn similar representations as the fixed PEs in the end, but in general, we'd rather want learned PEs to learn some representations that are better than the fixed ones.

> 3) enhanced generalization performance.

I do not see any real “enhanced generalization performance.” because (i) the evaluation on the test set is missing in all the experiments (performance is only reported on training and validation sets), and also crucially (ii) given that the authors refer, e.g., to Kazemnejad et al. 2023, I expected some length generalization experiments (which are much more challenging and interesting than in-domain generalization), e.g., LST with different grid sizes or fMRI with different brain sizes (for different individuals). Without such experiments, no strong generalization performance is demonstrated in the current paper, in my view.

**Other concerns:**

* I’m not convinced how broadly valid/applicable the reported claims are, e.g., the conclusions change when (i) regularization is additionally used, (ii) different optimizers are used, (iii) the results are rather noisy (e.g., compare 0.1 vs. 0.2 in Table A2; and 0.2 is better than 0.1).

* The unique takeaway message of Sec 4 is unclear to me.

* > However, it was surprising that c-nope models neither learned nor generalized

This is not surprising to me. There is a fundamental difference between the toy task used here, and the language modeling experiments used in previous works reporting positive results using c-NOPE: Unlike the language modeling task that is inherently auto-regressive (predicting the next token given all its predecessors), the LST task may require gathering information from all the positions to make a prediction. Please take a look at Figure 1.E. For example, when attending to position (2, 3), the left-to-right autoregressive model only has access to all positions up to (2, 3), while information from all other positions may be crucial to make a prediction. I expect this to largely hinder the model using a causal mask.

---

> ### Author Response · Authors · 2025-03-07
> **Author response to Reviewer jYDr (1/k)**
>
> We are grateful to the Reviewer for their careful and attentive review. Below, we work to summarize and address their concerns mentioned in the Weaknesses and Requested Changes.
>
> `1. Lack of connection to NTK theory, and ambiguity in choosing $\sigma$`
>
> We recognize that the current work primarily focuses on empirical evaluations. However, we would like to emphasize that the choices of $\sigma$ have been corroborated in prior empirical literature in simple feedforward networks (e.g., Flesch et al., 2022), and are also broadly consistent with those specified in theoretical papers (e.g., Chizat et al., 2020). Despite this, as alluded to by the Reviewer, specifics of what small means is likely dependent on the interplay of architecture and choice of optimizer, which can be challenging to assess from a theoretical perspective. NTK theory is derived from infinitely wide single hidden layer models, where gradients are straightforward to calculate.
> However, in practice, we suspect that the exact choice of $\sigma$ likely depends on the number of layers, the number of hidden units per layer, and the magnitude of the gradients that flow through these units.
> Though we agree that having such a connection between NTK theory and modern would be deeply insightful, we believe that this may be out of the scope of this paper; the present paper primarily focuses on empirical results of learning positional encodings for data organized in higher dimensions, and aims to demonstrate the broader utility of initialization dependent learning across a range of tasks and data types.
> However, in Response 6b below, we highlight some of the changes we have made to the Discussion that provide an in-depth discussion regarding how initialization likely interacts with optimization and gradient descent to determine the learning regime.
> Further, in future work, we hope to more rigorously address this theoretical connection, and thank the Reviewer for identifying this gap.

---

> > ### Author Response · Authors · 2025-03-07
> > **Author response to Reviewer jYDr (2/k)**
> >
> > `2. Using 1D PE baselines & learning representations/embeddings closely aligned to the ground truth.`
> >
> > We thank the Reviewer for pointing out this issue. We agree with the qualification that our results – namely that small-norm initializations outperform commonly-used *1D PE* baselines – is a more appropriate claim. *As such, we now refer to all instances of “relative” as “1d-relative” and “rope” as “1d-rope” in accordance with the Reviewer’s suggestion, and this is now reflected in the revised manuscript.* We do not intend to discredit the usage of other 2-D or *n*-D PEs, as we agree that other 2D PEs, such as a relative PE in 2D, would achieve similar success as the 2D sinusoidal PEs as long as they preserve position-wise isometry.
> >
> > Example of changes (p. 5); see also Fig. 5 and Table 1:
> > “We next sought to benchmark the optimal learnable PE (σ = 0.2, learn-0.2) relative to other standard PE schemes. These schemes included absolute 1D PE ($\texttt{1d-fixed}$, using sines and cosines; Vaswani et al. (2017)), 1D relative PE ($\texttt{1d-relative}$; Shaw et al. (2018)), and 1D rotary PE ($\texttt{1d-rope}$; Su et al. (2022)).
> >
> > Expanding on this point, we agree with the Reviewer that 2D PE is *not the only valid ground truth*. In reality, the ground truth positions for the Latin Squares Task is exactly the row and column information that corresponds to the 2D grid presented to humans (i.e., (x, y) coordinates, as visualized in Fig. 1).
> > Our rationale on why 2D PEs with sinusoids can be used to model the ground truth is because 1) it exactly preserves the original (x,y) coordinate information (i.e., a lossless mapping with respect to isometry between (x,y) and the 2D sinusoidal embedding (see Figure panels 1F v. 1H), and 2) it is a natural adaptation of the original (and still commonly-used) 1d sinusoidal PE from Vaswani et al. (2017) in 2 dimensions. In general, *any* PE that exactly preserves the pairwise distance between token PEs (e.g., Fig. 1F) can serve as a ground truth, including a simple PE such as $[x,y,0,…,0] \in \mathbb{R}^{d}$.
> > Indeed, this includes relative PEs in 2D that are isometric to Fig. 1F.
> > Note that Figures 1F and 1H are computed by measuring the pairwise distances between the embeddings between every pair of tokens. We have made clarifying changes to ensure that we acknowledge the possibility of many ground truths, so long as they preserve pairwise isometry of the original 2D grid.
> >
> > A key selection of updated texts:
> >
> > In the Abstract:
> > “Overall, we find that a learned PE initialized from a small-norm distribution can 1) uncover interpretable PEs that mirror ground truth positions **(with respect to isometry)** in multiple dimensions, and 2) lead to improved downstream generalization in empirical evaluations.”
> >
> > Caption of Figure 1F: “The pairwise distances between token positions according to rows and columns provides the ``ground truth'' of how tokens relate to each other in 2D space. *A successful PE would preserve the isometry (i.e., pairwise distance relationships) of the (x,y) grid coordinates.*”
> >
> > Section 3.2, p. 5: “(Note, that the term ``ground truth'' applies to any rotation of the absolute 2D PE, or any PE that preserves the pairwise isometry between the row and column information in the 2D LST grid. This could include alternative absolute 2D PEs not based on sinusoidals, in addition to relative PEs that preserve the isometry of the 2D grid information in the LST, as depicted in Fig. 1E,F).)”

---

> ### Author Response · Authors · 2025-03-07
> **Author response to Reviewer jYDr (3/k)**
>
> `3. Unwarranted claims of generalization.`
>
> Though we use the terminology of “training” and “validation” test sets, our use of this term is identical to “training” and “testing” sets, respectively. In particular, across all experiments, we measure generalization across independent train/test sets. In the human brain experiment, generalization is evaluated *across human participants*. This is a more conservative test of generalization then evaluating generalization on different time windows (or recording sessions) within the same individual. In the Latin Squares Task, all puzzles in the train/test set were distinct from those in the training set. Specifically, in the text we write:
>
> “We ensured that the similarity between any generated training set puzzle was distinct from the generalization (validation) set of puzzles (i.e., the Jaccard dissimilarity > 0.8 for a test puzzle to any individual training puzzle).” (p. 3).
>
> The Reviewer suggests “length generalization” as a more stringent test for generalization. While this would be interesting to assess, within the Latin Squares Task paradigm, it is not immediately feasible.
> This is because the rules of the task are to place a symbol in each square such that no symbol repeats in either a row or a column.
> This implies that the size of the grid scales with the number of unique symbols; a 4x4 grid requires exactly 4 unique symbols, and an $n$x$n$ grid requires exactly $n$ unique symbols.
> Thus, to test length (or size) generalization to greater grid sizes, the model would need to recognize and generalize to symbols that it has never seen before, which is difficult.
> In contrast, in 1D tasks (e.g., as elegantly studied in Kazemnejad et al., 2023), symbols commonly repeat, such as in arithmetic or context-free grammars.
>
> Nevertheless, to demonstrate that the generalization we test for here is non-trivial, we ran an experiment that shows that standard feedforward network cannot systematically generalize to the Latin Squares Task using the training and validation (testing) splits that we used here. This is in addition to the other experiments reported in the paper, in which we find that transformers with different positional encoding parameterizations exhibit varying degrees of generalization (e.g., $\texttt{1d-fixed}$ yields worse generalization than $\texttt{learn-0.2}$. These new results have now reported in the Appendix A16, which are referenced on p. 3:
>
> “Moreover, simpler neural network architectures, such as vanilla MLPs and bidirectional LSTMs, could not generalize well across this split (Fig. A16).”
>
> `4. Broad applicability of the reported findings due to differing results with hyperparameters.`
> > I’m not convinced how broadly valid/applicable the reported claims are, e.g., the conclusions change when (i) regularization is additionally used, (ii) different optimizers are used, (iii) the results are rather noisy (e.g., compare 0.1 vs. 0.2 in Table A2; and 0.2 is better than 0.1).
>
> Related to our response to comment #1, the focus of the present paper was to demonstrate the empirical relevance of initialization dependent PEs in transformers and their ability to learn interpretable and ground truth position embeddings (with respect to isometry). While regularization (weight decay) appears to widen the exact range of initialization norms that improve generalization, we find that the pattern remains the same – larger initializations lead to worse and slower generalization (despite immediate training convergence). Nevertheless, we provide additional nuance and discussion in the Discussion section on the impact of different optimizers and initializations in learning representations. The changes are copied below in response #6b.
>
> `5. What is the unique take-home message of Sec 4?`
>
> The takeaway from Section 4 is to demonstrate that the ability to recover ground truth network positions is dependent on the choice of PE initialization as a precursor to the fMRI experiment in the following section. For this, we used a toy network simulation where we explicitly chose the network positions before demonstrating in Fig. 4F that different PE initializations learn different ground truth PEs. In particular, the smaller the initialization, the closer the model’s PEs were to the defined networks. Our aim here was not to emphasize generalization, but learnability of ground truth positions, as determined by network clustering. We have added a sentence in the beginning of this section to improve its clarity (p. 7):
>
> “In our next experiment, we sought to demonstrate the impact of PE initialization in transformers in learning data representations from a nonlinear, stochastic network simulation. *Specifically, the main goal in this experiment was to assess the interpretability of learned representations, rather than generalization.*”

---

> ### Author Response · Authors · 2025-03-07
> **Author response to Reviewer jYDr (4/k)**
>
> `Regarding the statement: “However, it was surprising that c-nope models neither learned nor generalized”`
>
> We have revised this sentence to remove the ‘surprising nature’ of this result:
>
> “Poor performance was expected for the $\texttt{nope}$ model due to the lack of any explicit or implicit PE information, and $\texttt{c-nope}$ models neither learned nor generalized due to the autoregressive and causal structure of its attention mask.” (p. 5-6)
>
> `6. Requested Changes sections`
>
> `6a. Notations to positional encodings.`
>
> As mentioned above and requested by the Reviewer, we have changed the notations of $\texttt{relative}$ and $\texttt{rope}$ to $\texttt{1d-relative}$ and $\texttt{1d-rope}$, respectively.
> Though the Reviewer requested that we also evaluate the 2d versions of relative and rope, we feel that these additional experiments would not differ significantly from the existing results using $\texttt{2d-fixed}$ PE.
> This is because 1) $\texttt{2d-fixed}$ PE already performs at ceiling, and 2) the PE embeddings do not significantly differ with respect to isometry (i.e., the relative distances along a 2d grid, as visualized in Figures 1F and 1H).
> As stated above, the use of $\texttt{2d-fixed}$ PE as the ‘ground truth’ was to preserve the isometry of (x,y) coordinates of the 2D LST grid. Any PE that retains this isometry would also serve as a valid “ground truth” PE.
>
> `6b. Why 0.1 is the right order of magnitude relative to 0.01 and 0.001?`
>
> Prior studies have demonstrated that simple feedforward networks initialized from a small-distribution train very slowly due to small gradients, with some producing unrealistic training times (Flesch et al., 2022).
> Thus, while theory suggests that in the limit, small norm initializations should produce rich representations, in practice (and as previously suggested in empirical studies; Flesch et al. 2022), these can be hard to produce due to small gradients. These slower training times can be observed by the experiments we performed when using $\sigma=0.01$ and $\sigma=0.05$ (Figure A7), where slower training convergence is observed for SGD. What is interesting, however, is that when Adam is used, $\sigma=0.01$ leads to extremely poor generalization, which is putatively due to Adam's adaptive learning rates. Specifically, since the learning rate is scaled by the magnitude of the gradient flow in Adam, this behavior produces large changes to the weights (during the initial small gradient updates) that push the learning regime back into the lazy learning regime. We have now included a discussion of this to qualify the interpretation of our results, yet leave open the importance for more theoretical work on these differences on p. 12.
>
> “Finally, while we have empirically demonstrated that PE initialization impacts the learned representation and generalization in transformers, in practice, it remains unclear how the precise choice of $\sigma$ interacts with other hyperparameters (e.g., architectural choices and optimization protocols). For example, while theory alone suggests small norm initializations should produce rich representations, in practice, such initializations can be hard to train due to small gradients (Chizat et al., 2020). This can be empirically observed by the experiments we performed when using $\sigma$=0.01 and $\sigma$=0.05 (Fig. A4), where we find slower training convergence with smaller $\sigma$ using SGD. Interestingly, however, when Adam is used, $\sigma =0.01$ leads faster training convergence yet poor generalization, which is likely due to Adam's adaptive learning rates. Specifically, since the learning rate in Adam is scaled by the magnitude of the gradient flow in Adam, this behavior likely produces large changes to the weights (during the initial small gradient updates) that pushes learning regime back into the kernel learning regime. Nevertheless, it will be important for future theoretical work to more rigorously characterize these behaviors to better understand how the choice of $\sigma$, architecture, and optimizer influences representation learning.”
>
> `6c. Include experiments with $\sigma<0.1$`
>
> As mentioned above, we have included experiments with $\sigma$=0.05 and $\sigma$=0.01 in Figure A7 using both SGD and Adam optimizers.

---

> ### Author Response · Authors · 2025-03-07
> **Author response to Reviewer  jYDr (k/k)**
>
> `6d. Replace the toy experiment of Sec 4 with a new experiment with a ViT for image classification.`
>
> The motivation for Section 4 was to provide an experimental simulation to demonstrate the importance of PE initialization in learning interpretable positional (network) encodings in a stochastic simulation with known network clustering. The intention of this experiment was as a prelude to the experiments using real-world brain time series from fMRI data, and to demonstrate the extension of learning PEs beyond the 2D LST setup to learn interpretable positions in networked system (e.g., like the brain). While we greatly appreciate the Reviewer’s suggestion to demonstrate the impact of initialization in a more conventional 2D setting (e.g., image classification), we are uncertain as to what the additional experiment with a ViT for image classification would demonstrate beyond what was already shown with the 2D Latin Squares Task, since they are both 2D classification tasks. As such, though we are grateful for the suggestion, we have opted (for now) to not perform this experiment.
>
> `6e. Include test set experiments for generalization.`
>
> Kindly see our response in #3, “Claims of generalization”
>
> `6f. Regarding the claim “In neuroscience, a central goal is to be able to predict distributed brain activity using the activity of other brain regions” as being too narrow.`
>
> We have changed the phrasing of this from “a central goal” to “an important goal”.
>
> `6g. Typos`
>
> We have fixed the referenced typos – thank you for spotting them!
>
> Finally, thank you for taking the time to carefully evaluate our manuscript!

---

> ### Comment · Reviewer_jYDr · 2025-03-08
> **Thank you. Response acknowledged**
>
> I thank the authors for their response.
>
> However, the author’s response did not resolve any of my main concerns. Quoting the TMLR criteria, I am not convinced of the technical soundness, nor the clarity of the narrative and arguments presented.
>
> The author’s response confirms my original concern that some aspects of the theoretical framework appear to be either misinterpreted or not fully engaged with. In particular, one of the critical aspects of the NTK and related deep learning theories is to characterize learning regimes in terms of the "order" of dimensions defining the setting, e.g., scales of initial weights as a function of the dimensions of the weights. For example, in the very paper the authors cited in their response, Chizat et al., (2020) highlights the difference between O(1/m) vs O(1/sqrt(m)) scaling of the readout weight matrix where m is its output dimension (leading to rich vs lazy regimes, respectively).
>
> Here, since there is no theoretical foundation, the authors can arbitrarily define "what works" as "small", "what does not work" as "large", "what’s supposed to work but does not work in practice" as "very small" after observing the results (even though I sincerely thank the authors for these extra experiments; but for example, looking at Table A4, what would prevent me from concluding that within this range, a "small" sigma is a bad choice?).
>
> This is not technically sound, and I’m not convinced by the scientific values of superficial statements such as "**consistent with the NTK regime**, PE’s initialized from a distribution with large σ converged, but generalized poorly" (page 4) or "We noticed that for models that were initialized with very small-norm PEs, generalization performance was reduced, **counter to the theoretical claims in the NTK theory**" (Figure A7 caption).
>
> Returning to my original critique in the review, if I were to train a model that has an embedding size that is 10 times smaller (or larger) than the one used in this work, should I still initialize with sigma ~0.1? If not, what's the generalizable takeaway from this work? As I cannot find answer to these questions, to me the technical soundness of this work is limited to that of a collection of ablation studies.
>
> Regarding the experimental settings, having separate validation and test sets is crucial for evaluating generalization in machine learning. Hyper-parmeters (e.g. sigma in this work) can be selected based on the validation set performance, **but not the test set**. I regard this as a serious technical flaw.

---

> ### Author Response · Authors · 2025-03-10
> **Thank you for your acknowledgement**
>
> Thank you to the Reviewer for continuing to engage thoughtfully with our paper. We summarize the two continuing issues as 1) Generalizability of choice of sigma, and relation to the theoretical NTK framework, and 2) Partitioning of training and testing sets.
>
> `1) Generalizability of choice of $\sigma$ and relation to NTK`
>
> We acknowledge that our results are principally empirical in nature. We wish to highlight some differences of our experimental setup that differ from the NTK theory that provide evidence that our results are independent of scaling across embedding dimensions.
>
> The Reviewer is correct in that scaling matters in Chizat et al. (2020), when scaling of $O(1/m)$ vs. $O(1/\sqrt{m})$ can induce different rich v. lazy learning regimes. However, one distinguishing factor with our work is that Chizat et al. (2020) scales the initialization of the weight parameters (e.g., from input to hidden weights), where $m$ is the number of the downstream hidden units. In our work, we initialize exclusively the PE layer of transformers, which does not have a ‘read-out’ dimension (e.g, $m$ in the case of Chizat et al.), since PE vectors are superimposed on each token’s embedding. Moreover, standard sinusoidal PEs are scaled between 0 and 1 (i.e., bound by the range of a sines and cosines), and are invariant to the number of embedding dimensions. In other words, absolute sinusoidal positions are not scaled by the number of embedding dimensions.
> Thus, our scaling factor of $N(0,\sigma)$ is likely to be generalizable, **though we temper this claim in the Discussion in accordance with the Reviewer’s concerns, given the empirical nature of this finding**.
>
> Empirically we find support for this. Across our experimental findings (i.e., Latin Squares Task and human brain data), we find consistent evidence that the choice of $\sigma=0.1$ appears to be appropriate. In particular, we consistently find that the choice of $\sigma=0.1$ appears to induce rich learning in both of the Latin Squares Task and human fMRI data, despite 1) *different number of embedding dimensions in these two experiments ($d=160$ for the Latin Squares, and $d=64$ for brain data)*, and 2) different choices of loss function for each experiment (CrossEntropy for Latin Squres, and Mean Squared Error for human brain data. Importantly, choice of $\sigma$ is not scaled according to the number of embedding dimensions, which is consistent with the lack of scaling when using standard sinusoidal positional encodings.
>
> As mentioned in our prior response, we have caveated the results we find in Fig A7 that are “counter to theoretical claims in the NTK theory” in the revised version of the Discussion, where we 1) acknowledge the current limitations of the present study, and 2) provide an interpretation for why we see empirical results that contrast with NTK theory. To summarize, the contrasting results with NTK theory are likely to do with the assumption in Chizat et al. to use fixed gradient updates from “… stochastic gradient descent with small enough step sizes” (p. 4, Chizat et al. 2020), while in practice when using Adam, step sizes are adaptive and inversely related to the size of the gradient. (These interpretations are mentioned in the revised Discussion.)
>
> `2) Regarding generalization, and the technical flaw of not using both testing and validation sets.`
>
> We agree that if we were tuning a model or choosing hyperparameters to evaluate on a downstream test set, having separate test and validation sets would be important (or evaluating via cross-validation). This would be doubly important if we were evaluating the generalization performance of a single model (e.g., $\sigma$). However, in our experimental setting, *no hyperparameters were optimized on the test set*. Moreover, because $\sigma$ was the variable of interest (rather than a hyperparameter), our primary interest to assess generalization performance as a function of $\sigma$. Additionally, our results are reproduced and consistent across 15 independent model seeds per $\sigma$. Because we were explicitly interested in studying the relationship between generalization vs. $\sigma$, we respectfully disagree that this is a technical flaw.
>
> Again, thank you for thoughtfully engaging in our work!

---

> ### Comment · Reviewer_jYDr · 2025-03-10
> **This raises more concerns than it resolves**
>
> I appreciate the authors for their response; however, it raises more concerns regarding their scientific rigor and logic than it addresses.
>
> > despite 1) different number of embedding dimensions in these two experiments (d=160 for the Latin Squares, and d=64 for brain data)
>
> 160 and 64 are of the same order (there is only a factor of 2.5 in between). Finding that ~0.1 works in both cases does not allow us to conclude on its general applicability. If the authors really want to test the hypothesis that sigma is independent of the embedding size, it will at least require many more data points, specifically those that increase by factors of 10, and certainly across multiple model configurations (e.g., depth).
>
> Regarding the necessity of a validation and a separate test set, the authors are free to call sigma a "variable of interest" (or introduce any terminology they like), but it is fundamentally just a hyper-parameter in the language of machine learning. A sweep can be conducted to find its optimal value by monitoring the validation performance, but a separate test set is needed to evaluate generalization (simply because even hyper-parameter optimization may lead to overfitting to the validation set).
> The claim that "no hyperparameters were optimized on the test set" misrepresents the experimental setting, since drawing conclusions about generalization performance from a validation set is problematic; it effectively turns the validation set into a test set. Referring to a "downstream task" etc. is completely irrelevant to this discussion. This is a fundamental rule for rigorous machine learning.

---

> > ### Author Response · Authors · 2025-03-11
> > **Reply**
> >
> > `Regarding the first point, that our study lacks scientific rigor and logic:`
> >
> > We have mentioned above that we have tempered our claims and have provided speculative interpretation regarding the present empirical results in the Discussion. The Reviewer has appeared to disregard the additional explanation that prior uses of absolute PEs have similarly not been scaled by the number of embedding dimensions, and that our experiments follow this standard.
> >
> > `Regarding the second point on including test and validation sets: `
> >
> > The Reviewer claims that "even hyperparameter optimization may lead to overfitting to the validation set". Again, we emphasize that no model parameters or hyperparameters were optimized on the test set; all model performance on the validation set is independent, so no validation set overfitting is possible. We demonstrated this across 3 independent experiments with their own unique test sets, and reproduced across 15 random seeds per $\sigma$. Note that prior papers have also followed this standard (i.e., a single independent test set) when evaluating for rich and lazy learning (e.g., Chizat et al. 2020).

---

### Review · Reviewer_G9m9 · 2025-02-24

**Summary Of Contributions:**

The paper studies the transformer's ability to learn spatial relationships in the data.

They show that learnable positional encodings initialized with small norms generalize best on 3 tasks (1) Latin Square Task, (2) Masked prediction of a nonlinear stochastic network simulation, and (3) Masked prediction of spontaneous brain imaging data.

They limit themselves to small transformer encoders.

**Audience:**

Yes

**Claims And Evidence:**

Yes

**Requested Changes:**

> We note that choosing the initialization rank can also induce rich versus learning learning

Typo, s/learning learning/lazy learning

It may interesting to look at the learning dynamics to see if there is a more theoretical reason for the results.

**Strengths And Weaknesses:**

The experiments are convincing and well done. The data are different and intrinsic spatial relationships. Moreover, they benchmark against several other positional encoding schemes.

The main weakness are the results are mostly empirical. It's not clear why smaller norms work better. They also mention using L2-regularization without much justification.

---

> ### Author Response · Authors · 2025-03-07
> **Author response to Reviewer G9m9**
>
> We thank the Reviewer for their overall positive assessment of our manuscript. Below, we highlight some of the changes we have made to address some of the Reviewer's comments, as well as point to additional figures and results that we hope address some other concerns.
>
> `1. Weakness: Regarding the empirical nature of the paper:`
>
> Our broad motivation was to demonstrate the extremely strong inductive bias that can be induced by an ill-specified positional encoding choice for transformers, and provide alternative approaches to learning positions, particularly for tasks that have non-trivial or difficult to know positional encodings (e.g., in physically-embedded or networked systems, such as the brain or genetic networks). Additionally, we mention the impact of L2-regularization on learning these representations, as it is a common regularizer (e.g., weight decay) in modern systems. We have now clarified this in the text (p. 6):
>
> “In addition, we assessed the interaction of PE initialization with weight decay (AdamW), a commonly-used L2 regularization scheme (Loshchilov and Hutter, 2019). We empirically found that the combination of low initialization and weight decay improved generalization; performance was virtually indistinguishable from the ground truth 2d-fixed PE (both models generalized with 99% accuracy; Table A6; Fig. A8).”
>
> We have also provided more details onto the Discussion section, where we highlight the limitations of this paper (i.e., that it is primarily empirical), while alluding to some potential interesting theoretical directions to pursue for future work. We also provide intuition as to why certain values of $\sigma$ provide better representation learning with SGD vs. Adam.
>
> "Finally, while we have empirically demonstrated that PE initialization impacts the learned representation and generalization in transformers, in practice, it remains unclear how the precise choice of $\sigma$ interacts with other hyperparameters (e.g., architectural choices and optimization protocols). For example, while theory alone suggests small norm initializations should produce rich representations, in practice, such initializations can be hard to train due to small gradients (Chizat et al., 2020). This can be empirically observed by the experiments we performed when using $\sigma$=0.01 and $\sigma$=0.05 (Fig. A4), where we find slower training convergence with smaller $\sigma$ using SGD. Interestingly, however, when Adam is used, $\sigma =0.01$ leads faster training convergence yet poor generalization, which is likely due to Adam's adaptive learning rates. Specifically, since the learning rate in Adam is scaled by the magnitude of the gradient flow in Adam, this behavior likely produces large changes to the weights (during the initial small gradient updates) that pushes learning regime back into the kernel learning regime. Nevertheless, it will be important for future theoretical work to more rigorously characterize these behaviors to better understand how the choice of $\sigma$, architecture, and optimizer interact to influence representation learning.”
>
> `2. Typos`
>
> Thank you for identifying the typo – we have fixed it.
>
> `3. Show learning dynamics`
>
> We include plots of performance across training for the Latin Squares task in Figure 2B, and training loss for MSE on fMRI data in Figure 5c. We report the full training performances of the LST in Figure A8 with weight decay, in addition to the training trajectories across SGD and Adam in A7. Intuitively, very small initializations trained with SGD reveal significantly slower learning likely due to smaller gradients, which is consistent with prior papers and theory (Flesch et al., 2022). In contrast, small initializations with Adam do not show slower convergence, due to its adaptive learning rate (Fig. A7). However, and as mentioned above, extremely small model initializations trained with Adam (i.e., $\sigma$<0.1) yield poor generalization (see Fig. A7B), likely due to the adaptive learning rate, which scales the learning rate inversely to the magnitude of the gradient flow. This likely pushes the learning regime from the ‘rich’ training regime, back to the ‘lazy’ training regime. While prior work in neural tangent kernels have provided some theoretical intuition as why these initializations impact generalization (Chizat et al., 2020; Kunin et al., 2024), we hope to provide a better intuition of this work in transformers for future work.
>
> We thank the Reviewer for taking the time to review our paper!

---

### Decision · Action_Editor_ypuU · 2025-04-15

**Recommendation:** Reject

**Comment:**

Two out of the three reviewers felt that the evidence provided was sufficiently convincing to warrant a 'leaning accept', but the third reviewer was very clear that they felt the paper did not meet the bar of technical soundness required for publication in TMLR. After reading the reviews and author rebuttals, I was convinced that the critical reviewer had a point: the authors seem to select fairly arbitrary values for a variety of hyperparameters in the networks (dimensionality, depth, learning rate, etc.), and by their own admission they didn't do any hyperparameter tuning. Moreover, their choices for values of sigma weren't really rooted in any clear theory or empirical backing, making the lack of exploration of hyperparameters and broader values for sigma potentially more problematic. Given these considerations, it is not clear to me that these results would generalize. What if the best sigma was different under a different hyperparameter regime? There can be major interactions between the initialization of a network and other aspects of a network's design. As well, more direct connection to NTK would help reinforce the soundness of the results.

Given these considerations, I do not think this paper is appropriate for publication in TMLR as is. However, as noted by the other two reviewers, this paper could be of interest to TMLR readers, and moreover, the problems described above are fixable with more work. As such, I am leaving open the option for resubmission of a major revision at a later time. If the authors can address the concerns around generalizability, and run proper searches and/or greater exploration of hyperparameters (or improve connections to theory), then I think this paper could be appropriate for publication at TMLR.

**Audience:**

Yes, the TMLR audience, particularly those interested in using transformers outside of the language domain, would find this paper interesting.

**Claims And Evidence:**

This paper examines positional encodings (PE) for tasks with input structures beyond simple 1D sequences. The authors claim to show that the initialization of learnable PEs plays a critical role in capturing meaningful positional patterns and enhancing generalization. Across three tasks—including 2D reasoning, network simulation, and neuroscience data—the authors report that using a smaller sigma for the initialization leads to interpretable and effective PEs, improving both performance and model interpretability. They relate these findings to neural tangent kernel (NTK) theory, and argue that the results underscore the importance of PE design in non-trivial input domains.

The authors' empirical evidence does largely support their claims, with experiments showing the impact of PE initialization in a variety of interesting domains (specifically, the Latin Square Task, a synthetic time series, and masked prediction of brain imaging data). However, the connection to theory is impoverished, and there is a lack of appropriate practices with respect to verifying model performance (e.g. there was no split between validation and test data, and hyperparameters were not tuned), which weaken the support for the claims.

**Resubmission Of Major Revision:**

The authors may consider submitting a major revision at a later time.